# MapUQ: Map with Uncertainty Quantification for Robust BEV Vectorized Construction

**Shaoyuan Mo** [* 1]  **Qi Ma** [* 1]  **Rui Leng** [1]  **Bohan Li** [1]  **Ke Wang** [1]

## Abstract

End-to-end online map generation is a core component of autonomous driving perception systems. However, in complex traffic scenes, Bird's-Eye-View (BEV) with vectorized mapping suffers from limitations such as target misclassification, spatial localization drift, and ambiguous semantic segmentation. Introducing uncertainty quantification can alleviate these problems, so we propose MapUQ, a robust BEV vectorized mapping method guided by uncertainty-aware optimization. Specifically, we quantify uncertainty at the feature level to enhance semantic perception, apply an error-driven dynamic receptive field adaptation mechanism at the decoding stage to enforce geometric consistency, and leverage negative sample information at the output head to improve lane classification accuracy. Experimental results on the nuScenes and Argoverse 2 datasets show that our method outperforms prior approaches in AP across three road types, achieving an average improvement of 1.5% over the baseline with marginal computational overhead. In addition, our method surpasses the baseline on uncertainty metrics such as ECE and NLL, significantly improving robustness and mapping accuracy in complex scenarios. Our code has been released at github: https://github.com/CQU-AVL/MapUQ

## 1. Introduction

Recently, numerous Bird's-Eye-View (BEV)-based online mapping methods have achieved remarkable performance leveraging Transformers (Vaswani et al., 2017; Liu et al., 2023; Liao et al., 2023; 2025; Qiao et al., 2023). BEV vectorized mapping models road elements as sets of instance points with topological structures (Wu et al., 2024), unifying multi-view sensor data into a physically consistent representation (Liu et al., 2023; Li et al., 2022a). Key sub-tasks of BEV mapping are generally categorized into three types: object classification (Liao et al., 2023; Li et al., 2022b), spatial localization (Huang et al., 2021; Xie et al., 2022), and semantic segmentation (Zhou & Krähenbühl, 2022; Peng et al., 2023).

Compared to traditional LiDAR schemes, vision-only BEV vectorized mapping offers simpler deployment and cost advantages (Hao et al., 2025; Zhang et al., 2024b). However, these methods encounter notable limitations in complex real-world scenarios(Monninger et al., 2025). First, severe occlusion disrupts spatial consistency, degrading spatial localization precision (Xie et al., 2022; Roddick & Cipolla, 2020). Second, complex road conditions complicate the recognition of rare landmarks, impacting object classification (Li et al., 2023; Wang et al., 2023). Third, in extreme weather, reduced contrast leads to erroneous predictions in semantic segmentation (Yu et al., 2024; Li et al., 2022c). To mitigate these risks, incorporating uncertainty quantification mechanisms is critical.

While uncertainty quantification methods like MC-Dropout (Gal & Ghahramani, 2016) are applied in 2D tasks, their application in Transformer-based BEV vectorization remains unexplored. Moreover, traditional uncertainty methods often require multiple forward passes (Gal & Ghahramani, 2016; Lakshminarayanan et al., 2017; Kendall & Gal, 2017; Blundell et al., 2015), which is computationally prohibitive for real-time autonomous driving. To address this, we propose MapUQ, a BEV vectorized map based on uncertainty quantification. Specifically, the Multi-task Semantic Uncertainty Head (MSUH) quantifies uncertainty within BEV features via MC-Dropout, distinguishing between aleatoric and epistemic uncertainty to provide robust priors. The Progressive ROI Scale Adapter (PRSA) dynamically generates scaling factors based on geometric deviations to modulate the Deformable Attention search range, preventing polyline fragmentation. Finally, the Negative Sample Classifier (NSC) integrates uncertainty scores into a hard negative mining strategy, granting the model a self-inspection capability for spurious predictions.

---

[*]Equal contribution  [1]Chongqing University, College of Mechanical and Vehicle Engineering, Chongqing, China. Correspondence to: Ke Wang <kewang@cqu.edu.cn>.

*Proceedings of the 43rd International Conference on Machine Learning*, Seoul, South Korea. PMLR 306, 2026. Copyright 2026 by the author(s).

In summary, the main contributions of this paper are as follows:

- We integrate uncertainty quantification into the transformer-based vectorized BEV mapping framework. This enables the autonomous driving system to generate more trustworthy vectorized mapping results in complex and unseen scenarios, significantly enhancing model robustness and reliability.

- We propose three novel modules to embed uncertainty awareness into the BEV vectorization pipeline: the Multi-task Semantic Uncertainty Head (MSUH), the Progressive ROI Scale Adapter (PRSA), and the Negative Sample Classifier (NSC). These components collectively improve detection performance and output reliability.

- We conduct extensive experiments on the nuScenes and Argoverse 2 datasets, including performance benchmarking, ablation studies, and visualization analysis. In addition, we conducted experimental evaluations on uncertainty quantification for complex scenarios. Experimental results demonstrate that our method yields superior mapping quality in challenging environments, advancing the application of uncertainty quantification in autonomous driving.

## 2. Related Works

**BEV-based Online Mapping**. Leveraging robust spatial representation, these algorithms are central to HD map construction (Zhu et al., 2023; Jiang et al., 2023). HDMap-Net (Li et al., 2022a) pioneered this by formulating mapping as semantic segmentation; however, it lacked topological connectivity and required heavy post-processing. Consequently, research shifted toward end-to-end vectorization. VectorMapNet (Liu et al., 2023) modeled map elements as point sequences using auto-regressive Transformers. MapTR (Liao et al., 2023) significantly accelerated inference via permutation-equivalent modeling and hierarchical queries. Recent works focus on geometric and temporal optimization: BeMapNet (Qiao et al., 2023) utilizes Bézier curve fitting for smooth lanes; PivotNet (Ding et al., 2023) leverages keypoint detection for complex topologies; and StreamMapNet (Yuan et al., 2024) introduces temporal fusion to mitigate occlusion-induced deficits.

MapTRv2 (Liao et al., 2025) achieves state-of-the-art (SOTA) performance on nuScenes (Caesar et al., 2020) by integrating hierarchical queries and multi-scale feature sampling. Due to its balance between geometric accuracy and convergence speed, we adopt MapTRv2 as the baseline framework for this study.

**Uncertainty Quantification (UQ)**. UQ is essential for handling perceptual noise (Nayak et al., 2024; Wang et al., 2025a). Mainstream approaches include MC-Dropout (Gal & Ghahramani, 2016) and Deep Ensembles (Fort et al., 2019). MC-Dropout (Gal & Ghahramani, 2016) approximates Bayesian inference via regularization, while Deep Ensembles (Fort et al., 2019) improve stability through independent networks. Despite their effectiveness, these methods require multiple forward passes or significant memory, rendering them computationally prohibitive for real-time industrial mapping tasks (Bui & Liu, 2024; Li et al., 2024).

In contrast, distance-based methods (Kanamori et al., 2009; Nguyen et al., 2022; Van Amersfoort et al., 2020; Feng & Wang, 2024) offer lightweight, single-pass alternatives for modeling feature space errors. Specifically, Mahalanobis distance-based deterministic quantification effectively captures distributional shifts while maintaining efficiency (Zhang et al., 2024a; Lv et al., 2025). Consequently, our proposed MapUQ integrates Mahalanobis distance quantification with optimized MC-Dropout to refine the BEV architecture, ensuring reliability in complex autonomous driving scenarios.

## 3. Method

### 3.1. Overview of the Algorithm Framework

Building upon the analysis of uncertainty in online vectorized mapping, we propose MapUQ, a robust framework grounded in the Transformer-based BEV architecture that integrates uncertainty quantification across the entire pipeline. As illustrated in Figure 1, MapUQ comprises three core modules designed to address uncertainty at different processing stages: the Multi-task Semantic Uncertainty Head (MSUH) operates at the feature level to quantify aleatoric and epistemic uncertainty via MC-Dropout; the Progressive ROI Scale Adapter (PRSA) dynamically modulates the Deformable Attention (Sun et al., 2025) search range within decoder layers based on geometric prediction errors; and the Negative Sample Classifier (NSC) leverages uncertainty scores at the final output stage to execute hard negative mining, effectively suppressing spurious detections in complex environments.

### 3.2. Multi-task Semantic Uncertainty Head (MSUH)

Consider the BEV spatial feature $f \in \mathbb{R}^{h \times w \times c}$, where $h$, $w$, and $c$ denote the height, width, and channel count of the feature map, respectively. Specifically, by maintaining Dropout activation during both training and inference phases, the model performs $T$ stochastic forward passes on the same feature representation. This process constitutes sampling within the model's hypothesis space to evaluate the consistency of predictions across different sub-network structures for a fixed input. For each sampling step $t \in \{1, \cdots, T\}$, we ob-

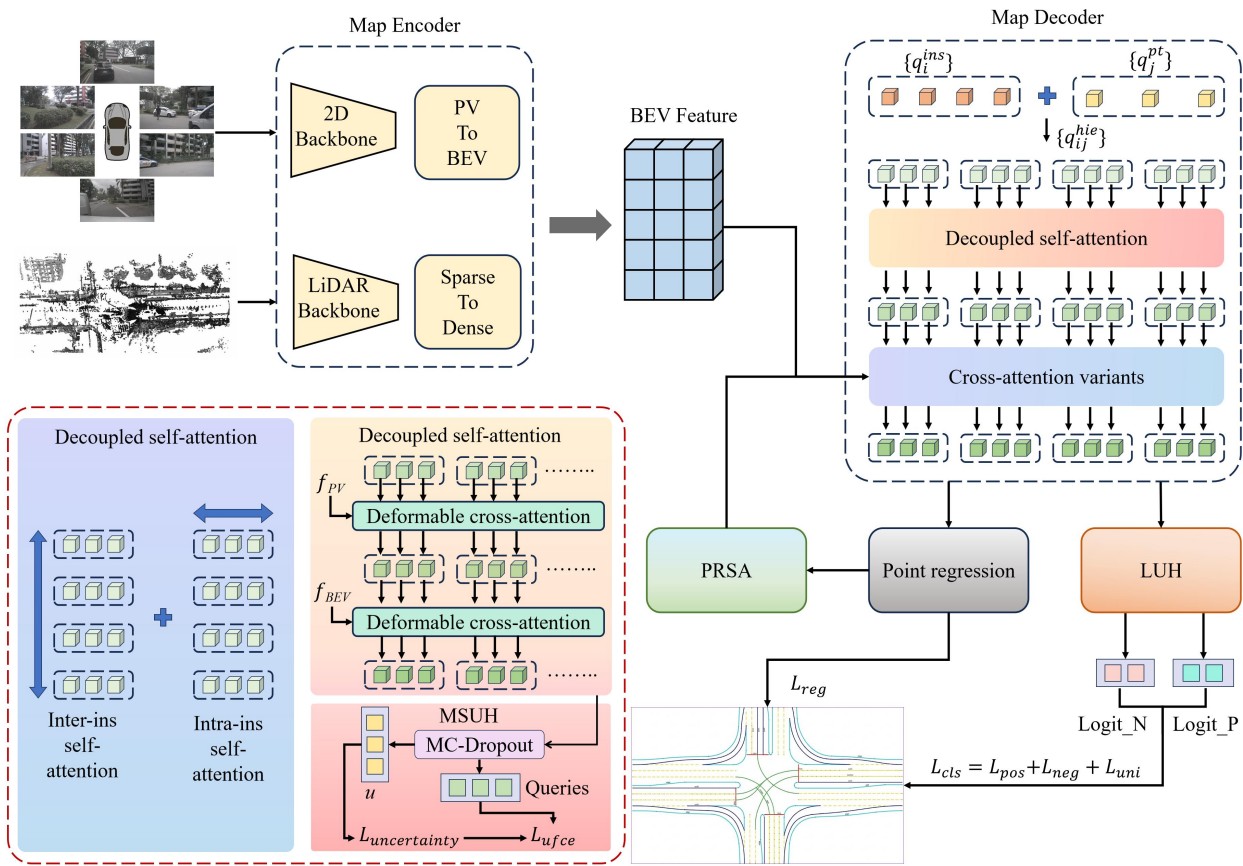

*Figure 1.* Framework of the proposed MapUQ. MSUH represents Multi-task Semantic Uncertainty Head; PRSA represents the Uncertainty-driven Progressive ROI Scale Adapter; NSC represents Negative Sample Classifier.

tain prediction logits $\hat{y}_t$ via random neuronal dropout. These auxiliary segmentation logits encode rich information regarding the semantic probability distribution. Subsequently, we apply the Softmax function to these probability values and subtract the average output probability distribution $\bar{p}$ to obtain the average prediction probability distribution. By aggregating the results from these $T$ trials, MSUH derives two critical uncertainty metrics (Wang et al., 2025b):

epistemic uncertainty:

$$U_{epi} = \frac{1}{T} \sum_{t=1}^{T} \left(\text{Softmax}(\hat{y}_t) - \bar{p}\right)^2 \tag{1}$$

aleatoric uncertainty:

$$U_{ale} = -\sum_{k=1}^{K} \bar{p} \log(\bar{p} + \epsilon) \tag{2}$$

where $\epsilon$ represents a numerical stability term.

Building upon the uncertainty prediction head, we design a task-adaptive uncertainty-aware loss framework to fully leverage uncertainty information during supervised learning. Specifically, for the boundary semantic segmentation task, we introduce an uncertainty-based dynamic weighting mechanism atop the standard pixel-wise cross-entropy loss. The segmentation loss is weighted by the magnitude of the uncertainty, where the weighting term is given by:

$$\omega_i = 1 + \alpha \cdot \hat{U}_{epi}(i) + \beta \cdot \hat{U}_{ale}(i) \tag{3}$$

where $\hat{U}$ denotes the normalized uncertainty value, and $\alpha$ and $\beta$ represent the uncertainty weighting coefficients, and $i$ represents the BEV grid location (pixel index). Consequently, the final weighted segmentation loss is defined as:

$$\mathcal{L}_{wCE} = -\frac{1}{|\Omega|} \sum_{i \in \Omega} \omega_i \cdot \log p_{y_i}(x_i) \tag{4}$$

where $\Omega$ denotes the set of valid samples.

Furthermore, to prevent the generation of excessive invalid uncertainty, we introduce an uncertainty regularization term:

$$\mathcal{L}_{\text{unc-reg}} = \frac{1}{|\Omega|} \sum_{i \in \Omega} \left(U_{\text{epi}}(i) + U_{\text{ale}}(i)\right) \tag{5}$$

To enhance predictive stability, we further enforce consistency across different stochastic MC Dropout samples. Specifically, we define a consistency loss formulated as the Kullback-Leibler (KL) divergence:

$$\mathcal{L}_{MC} = \frac{1}{T(T-1)} \sum_{t \neq s} \text{KL}\left(p^{(t)} \| p^{(s)}\right) \qquad (6)$$

where $p^{(t)}$ and $p^{(s)}$ denote the predictive probability distributions obtained from the $t$-th and $s$-th stochastic MC Dropout forward passes, respectively. This constraint effectively suppresses predictive oscillations in high-uncertainty regions.

To enhance boundary discrimination, we partition pixels into high-confidence and low-confidence sets based on their predictive confidence, and introduce a contrastive constraint:

$$\mathcal{L}_{\text{contrast}} = -\log\left(\exp(m) + \mathbb{E}\left[\langle \boldsymbol{p}_h, \boldsymbol{p}_l \rangle\right]\right) \qquad (7)$$

where $p_h$ and $p_l$ represent the predictive probabilities associated with the high-confidence and low-confidence prediction sets, respectively. Additionally, $m$ denotes the contrastive margin, designed to separate the feature distributions of predictions with different confidence levels.

Combining the aforementioned components, the final loss for the boundary-level segmentation task is defined as:

$$\mathcal{L}_{\text{seg}} = \mathcal{L}_{wCE} + \lambda_{\text{unc}}\mathcal{L}_{\text{unc-reg}} + \lambda_{MC}\mathcal{L}_{MC} + \lambda_{\text{con}}\mathcal{L}_{\text{contrast}} \qquad (8)$$

where $\lambda_{unc}$, $\lambda_{MC}$, and $\lambda_{con}$ are weighting coefficients that balance the relative contributions of the individual loss terms.

### 3.3. Uncertainty-driven Progressive ROI Scale Adapter (PRSA)

To address the issue of vectorization fragmentation caused by fixed receptive fields in complex scenarios (Goo et al., 2025), we propose the Progressive ROI Scale Adapter (PRSA), which dynamically modulates spatial search boundaries via a closed-loop feedback mechanism based on geometric uncertainty, as shown in Figure 2. Formally, we compute the geometric uncertainty as the $L_2$ error norm, denoted as $err$, between the predicted points $P_{pred}$ and the ground truth $P_{gt}$:

$$err = \|P_{pred} - P_{gt}\|_2 \qquad (9)$$

In the inference phase, where ground truth is unavailable, we utilize the dispersion of the predicted point sequence as a proxy metric for uncertainty. Based on this error (or proxy), PRSA computes an adaptive scaling factor, $S_{factor}$, for the subsequent decoder layer:

$$S_{\text{factor}} = 1.0 + \gamma \cdot \frac{err}{\tau + \epsilon} \qquad (10)$$

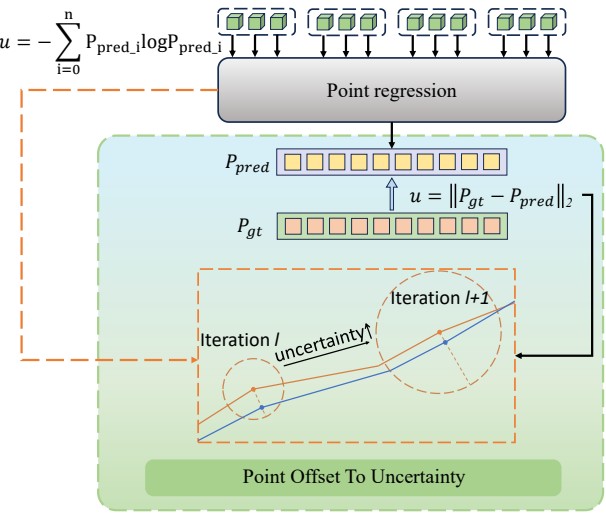

*Figure 2.* Framework of the proposed PRSA. The orange arrows denote the execution path during inference.

where $\gamma$ denotes the scaling intensity coefficient, and $\text{Clamp}$ is the truncation constraint function. Consequently, the ROI scales for the next layer, denoted as $Scales_{next}$, are updated as follows:

$$\text{Scales}_{\text{next}} = \text{Clamp}\left(\text{Scales}_{\text{prev}} \times S_{\text{factor}}, \ S_{\text{min}}, \ S_{\text{max}}\right) \qquad (11)$$

where $S_{min}$ and $S_{max}$ represent the lower and upper bounds of the ROI scales, respectively, and $Scales_{prev}$ denotes the ROI scales from the preceding layer.

Specifically, when predictive uncertainty is high, the model automatically expands its receptive field to capture broader contextual information, thereby effectively preventing the regression fragmentation of polylines.

While uncertainty-driven scale adaptation enhances robustness, unconstrained adaptation may lead to severe scale oscillations between decoding layers. To address this, we introduce an ROI scale consistency loss to regularize the scale evolution process. Let $|Q|$ denote the total number of queries, and $q \in \{1, \dots, |Q|\}$ represent the lane query index. The scale consistency loss is defined as:

$$\mathcal{L}_{\text{scale-cons}} = \frac{1}{|\mathcal{Q}|} \sum_{q=1}^{|\mathcal{Q}|} \left\| \text{Scale}_{\text{next}}^{(q)} - \text{Scale}_{\text{prev}}^{(q)} \right\|_1 \qquad (12)$$

Furthermore, based on the error-aware target scale constructed within PRSA:

$$\text{Scales}_{\text{target}}^{(q)} = \text{Scales}_{\text{prev}}^{(q)} \times \text{Scales}_{\text{factor}}^{(q)} \qquad (13)$$

the scale regularization term can be reformulated as:

$$\mathcal{L}_{\text{scale-target}} = \frac{1}{|\mathcal{Q}|} \sum_{q=1}^{|\mathcal{Q}|} \left\| \text{Scale}_{\text{next}}^{(q)} - \text{Scale}_{\text{target}}^{(q)} \right\|_1 \qquad (14)$$

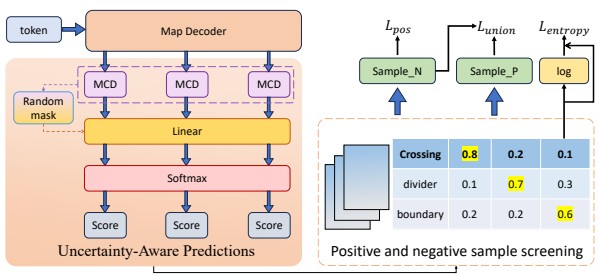

*Figure 3.* Framework of the proposed NSC.sample_P and sample_N represent the sets of obtained positive and negative samples, respectively.

This formulation explicitly aligns the scale updates with the geometric uncertainty, denoted as $err$, compelling the model to proactively expand the ROI search range in regions with high error.

Synthesizing the aforementioned constraints, the final ROI scale regularization term is defined as:

$$\mathcal{L}_{\text{ROI-scale}} = \lambda_{\text{scale}} \cdot \mathcal{L}_{\text{scale-}\frac{\text{cons}}{\text{target}}} \tag{15}$$

where $\lambda_{scale}$ represents the loss weighting coefficient. In practice, one may select either the scale-cons or scale-target formulation depending on the specific implementation requirements.

### 3.4. Negative Sample Classifier (NSC)

To mitigate spurious detections caused by model overconfidence (Aliferis & Simon, 2024), we propose the Negative Sample Classifier (NSC), which employs an uncertainty-based hard negative mining strategy to identify and penalize high-confidence outliers exhibiting anomalous feature representations, as shown in Figure 3. Specifically, we introduce MC Dropout into the classification head to perform $T$ stochastic forward passes for each query $q$. For the $t$-th sampling, the output logits $z_q^{(t)}$ are processed via Softmax:

$$p_q^{(t)} = \text{Softmax}(\hat{z}_q^{(t)}) \tag{16}$$

The set of stochastic predictions, denoted as $\{p_q^{(t)}\}_{t=1}^T$, probabilistically aligns with the segmentation branch, serving as a unified input for the subsequent lane uncertainty modeling and uncertainty-guided sample mining.

Leveraging these $T$ stochastic samples, we simultaneously model both epistemic uncertainty and aleatoric uncertainty from the predictive distribution. First, we compute the mean predictive probability under MC dropout:

$$\bar{p}_q = \frac{1}{T} \sum_{t=1}^T p_q^{(t)} \tag{17}$$

Epistemic uncertainty reflects uncertainty at the level of model parameters. In this work, we estimate it via the variance of the predicted probabilities across the MC sampling dimension:

$$U_{\text{epi}}(q) = \frac{1}{K} \sum_{k=1}^K \text{Var}_{t=1}^T \left( p_{q,k}^{(t)} \right) \tag{18}$$

where $p_{q,k}^{(t)}$ denotes the Softmax probability that the $q$-th lane query belongs to class $k$ during the $t$-th stochastic MC Dropout forward pass. Simultaneously, we utilize the entropy of the mean predictive distribution to characterize the overall predictive uncertainty, referred to here as aleatoric uncertainty:

$$U_{ale}(q) = -\sum_{k=1}^K \bar{p}_{q,k} \log \left( \bar{p}_{q,k} + \epsilon \right) \tag{19}$$

Upon obtaining uncertainty estimates, we leverage them for sample selection and negative sample utilization within the lane classification task. Predictions that are correctly classified and possess a maximum class probability exceeding a confidence threshold are identified as positive samples:

$$\mathcal{P} = \left\{ q \,\middle|\, \arg\max_k \bar{p}_{q,k} = y_q \wedge \max_k \bar{p}_{q,k} > \delta \right\} \tag{20}$$

where $y_q$ represents the ground truth class label for the $q$-th lane query, and $\delta$ denotes the confidence threshold for positive samples. Conversely, negative samples are automatically mined from high-uncertainty predictions. Specifically, based on the epistemic uncertainty $U_{epi}$, we select the top $r\%$ of predictions from the uncertainty distribution to form the negative sample set:

$$\mathcal{N} = \left\{ q \,|\, U_{epi}(q) > Quantile_r(U_{epi}) \right\} \tag{21}$$

To enhance discriminative capability in complex scenarios, we design an uncertainty-aware lane classification cross-entropy loss function that achieves robust optimization by jointly modeling predictive uncertainty, stochastic consistency constraints, and uncertainty-guided sample mining.

$$\mathcal{L}_{cls} = \frac{1}{|\Omega|} \sum_{q \in \Omega} CE \left( \bar{p}_q, y_q \right) \tag{22}$$

To econstrain the model's predictive stability under stochastic sampling, we regularize the classification results from the perspectives of both epistemic and aleatoric uncertainty:

$$\mathcal{L}_{epi} = \frac{1}{|\Omega|} \sum_{q \in \Omega} U_{epi}(q) \tag{23}$$

$$\mathcal{L}_{ale} = \frac{1}{|\Omega|} \sum_{q \in \Omega} U_{ale}(q) \tag{24}$$

Furthermore, to mitigate excessive stochastic fluctuations introduced by MC Dropout sampling, we impose consistency constraints on the results of different stochastic forward passes. Specifically, we employ the Kullback-Leibler (KL) divergence (Cui et al., 2024) between MC predictive probability distributions to quantify their discrepancy:

$$\mathcal{L}_{mc-cons} = \frac{2}{T(T-1)} \sum_{i<j} \frac{1}{|\Omega|} \sum_{q \in \Omega} KL\left(p_q^{(i)} \| p_q^{(j)}\right) \tag{25}$$

regarding sample-specific weighting, we assign higher weights to the classification loss for high-uncertainty negative samples to bolster the model's discriminative capability. Conversely, for high-confidence positive samples, we apply milder regularization constraints to maintain stable learning:

$$\mathcal{L}_{neg} = \frac{1}{|\mathcal{N}|} \sum_{q \in \mathcal{N}} CE(\bar{p}_q, y_q) \tag{26}$$

$$\mathcal{L}_{pos} = \frac{1}{|\mathcal{P}|} \sum_{q \in \mathcal{P}} CE(\bar{p}_q, y_q) \tag{27}$$

Synthesizing the above components, the final uncertainty-aware lane classification loss function is defined as:

$$\mathcal{L}_{lane} = \mathcal{L}_{cls} + \lambda_u(\mathcal{L}_{epi} + \mathcal{L}_{ale}) \\ + \lambda_{mc}\mathcal{L}_{mc-cons} + \lambda_{neg}\mathcal{L}_{neg} + \lambda_{pos}\mathcal{L}_{pos} \tag{28}$$

where $\lambda_u$, $\lambda_{mc}$, $\lambda_{neg}$, and $\lambda_{pos}$ are weighting coefficients used to balance the relative contributions of the individual loss terms.

## 4. Experiments

### 4.1. Dataset

In this chapter, we conduct training and evaluation on the well-known open-source autonomous driving dataset nuScenes (Caesar et al., 2020). Additionally, the experimental results on the Argoverse 2 dataset (Wilson et al., 2023) are presented in the appendix.

The nuScenes dataset comprises 1,000 complex urban driving scenes, each spanning approximately 20 seconds. It provides a comprehensive suite of sensor data alongside high-definition (HD) map annotations.The Argoverse 2 dataset consists of 1,000 driving sequences, with each sequence lasting 15 seconds. It features data from seven high-resolution surround-view cameras, accompanied by detailed 3D HD map annotations that encompass rich geometric and topological information, including lane centerlines, road boundaries, and pedestrian crossings.

### 4.2. Evaluation Criterion

We evaluate map construction quality using Average Precision (AP) across three categories and measure efficiency

via Frames Per Second (FPS). To rigorously assess uncertainty estimation, we employ Expected Calibration Error (ECE) and Negative Log Likelihood (NLL) for calibration, alongside AUSE, AUROC, and False Positive Rate (FPR) for effectiveness.

$$AP = \sum_{k=1}^{n} (r_{k+1} - r_k) \max_{\tilde{r} \geq r_{k+1}} p(\tilde{r}) \tag{29}$$

$$ECE = \sum_{m=1}^{M} \frac{|B_m|}{N} |acc(B_m) - conf(B_m)| \tag{30}$$

$$NLL = \frac{1}{N} \sum_{i=1}^{N} \left( \frac{\log(\hat{\sigma}_i^2)}{2} + \frac{(y_i - \hat{\mu}_i)^2}{2\hat{\sigma}_i^2} \right) + const \tag{31}$$

### 4.3. Implementation Details

We adopt ResNet-50 (He et al., 2016) as the backbone network and utilize the AdamW optimizer with a weight decay of $1 \times 10^{-4}$. The batch size is set to 32. The entire model is trained on four NVIDIA A100 GPUs for 24 epochs by default, with an initial learning rate of $6 \times 10^{-4}$. Regarding the module-specific hyperparameters, in the NSC module, the default number of negative samples is set to 2, with a discrimination threshold of 0.5 and a dropout rate of 0.1. For the PRSA module, we set the scaling intensity coefficient to 0.75, the temperature parameter to 1.5, and the scaling range is bounded within [0.7,1.5].

### 4.4. Main Result

Table 1 presents quantitative comparison on the nuScenes validation set. MapUQ consistently outperforms the baseline model, MapTRv2 (Liao et al., 2025), across various training configurations. Specifically, when trained with a ResNet-50 backbone for 110 epochs, our model achieves an mAP of 69.7%, representing a 1.0% improvement over the baseline. Notably, under the short-cycle training regime of 24 epochs, MapUQ demonstrates superior convergence properties, yielding an mAP gain of 1.5% (63.0% vs. 61.5%). We attribute this accelerated optimization to the NSC module, which leverages uncertainty to guide the mining of hard samples. Furthermore, the accuracy for the geometrically complex "pedestrian crossing" category improves by nearly 2%, validating the effectiveness of the MSUH module in perceiving ambiguous boundaries. This performance enhancement is achieved with a negligible compromise in inference speed (maintaining 13.9 FPS).

### 4.5. Ablation Study

Table. 2 presents the ablation study on NuScenes, evaluating how different uncertainty quantification (UQ) modules

*Table 1.* **Comparison with state-of-the-art methods on the NuScenes val set.** "ped.", "div." and "bou." represent pedestrian crossing, divider, and road boundary, respectively. **MapUQ** achieves the best performance under the same settings.

| Method | Backbone | Epoch | AP (%) | | | | FPS |
| --- | --- | --- | --- | --- | --- | --- | --- |
| | | | ped. | div. | bou. | mean | |
| VectorMapNet | R50 | 110 | 42.5 | 51.4 | 44.1 | 46.0 | 2.2 |
| BeMapNet | R50 | 30 | 62.3 | 57.7 | 59.4 | 59.8 | 8.3 |
| | R50 | 30 | 62.3 | 57.7 | 59.4 | 59.8 | 8.3 |
| | SwinT | 30 | 64.4 | 61.2 | 61.7 | 62.4 | 7.9 |
| MapVR | R50 | 110 | 55.0 | 61.8 | 59.4 | 58.8 | 15.1 |
| | R50 | 24 | 47.7 | 54.4 | 51.4 | 51.2 | 15.1 |
| | SwinT | 110 | 68.0 | 62.6 | 69.7 | 66.8 | 8.3 |
| PivotNet | SwinT | 30 | 63.8 | 58.7 | 64.9 | 62.5 | 8.3 |
| | R50 | 30 | 53.8 | 55.8 | 59.6 | 57.4 | 10.4 |
| MapTR | R18 | 110 | 39.6 | 49.9 | 48.2 | 45.9 | 35.0 |
| | R50 | 110 | 56.2 | 59.8 | 60.1 | 58.7 | 15.1 |
| | R50 | 24 | 46.3 | 51.5 | 53.1 | 50.3 | 15.1 |
| MapTRv2 | R18 | 110 | 48.1 | 55.9 | 55.6 | 53.2 | 32.5 |
| | R50 | 110 | 68.1 | 68.3 | 69.7 | 68.7 | 14.1 |
| | R50 | 24 | 59.8 | 62.4 | 62.4 | 61.5 | 14.1 |
| **MapUQ (Ours)** | R50 | 110 | **70.0** | **68.7** | **70.5** | **69.7** | 13.9 |
| | R50 | 24 | **62.3** | **63.2** | **63.6** | **63.0** | 13.9 |

impact the MapUQ model. The "Baseline" model yields a mean Average Precision (AP) of 61.5. Introducing individual UQ modules (Unc-cls, Unc-reg, Unc-seg in MapUQ_1, _2, _3 respectively) consistently provides slight improvements, with MapUQ_1 achieving 62.1 mean AP. Combining these modules further enhances performance; for instance, MapUQ_4 (Unc-cls + Unc-reg) reaches 62.3 mean AP, and MapUQ_5 (Unc-cls + Unc-seg) achieves 62.6. The most comprehensive model, MapUQ_7 (all three UQ modules), demonstrates the highest mean AP of 63.0. This indicates that explicitly modeling uncertainty in classification, regression, and segmentation effectively boosts perception accuracy on NuScenes, with negligible impact on inference speed (around 14.0-14.1 FPS).

### 4.6. Real-time Analysis

The computational cost analysis in Table 3 further corroborates the high efficiency of MapUQ. Within the total inference latency of 71.9 ms, the computationally intensive Map Encoder accounts for the majority of the runtime (75.10%). In contrast, the Output Head, which incorporates our core uncertainty estimation improvements, requires only 0.8 ms (constituting a mere 1.12%). This indicates that the additional computational overhead introduced by the uncertainty quantification mechanism is virtually negligible. MapUQ

successfully demonstrates that achieving high-precision and high-reliability vectorized mapping is entirely feasible without compromising real-time performance (14 FPS).

### 4.7. Uncertainty Evaluation

To assess the reliability of the predicted probability distributions and the efficacy of uncertainty estimates in indicating prediction errors, we conducted calibration and effectiveness evaluations on the nuScenes dataset, as detailed in Table 4. These results further substantiate the superiority of MapUQ in uncertainty estimation. Compared to the baseline model, which exhibits severe over-confidence (ECE = 13.1), our method reduces the ECE to 9.3, achieving significantly improved confidence calibration. Crucially, the model demonstrates exceptional failure prediction capability (achieving an AUROC of 91.8%) and leverages uncertainty information to suppress the False Positive (FP) Rate from 15.7% to 8.4%. These findings indicate that MapUQ not only generates accurate maps but also possesses the capability of "knowing when it might fail," thereby substantially enhancing the safety of the autonomous driving system.

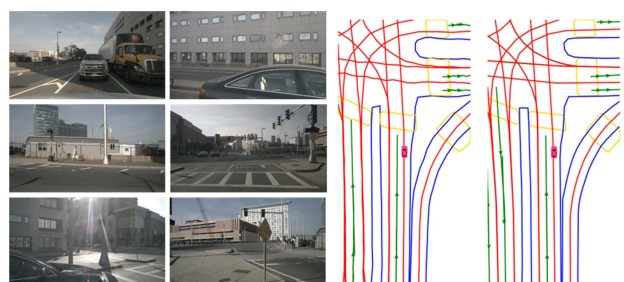

*Figure 4.* Visualization example in NuScenes dataset.

### 4.8. Visual Analysis

#### 4.8.1. RECONSTRUCTION QUALITY ON NUSCENES

Fig.4 illustrates the reconstruction quality on the NuScenes dataset. Faced with dense parallel lanes and complex intersection topologies, MapUQ generates vectorized lines with exceptional geometric precision and smoothness. Thanks to the dynamic ROI adjustment in the PRSA module, the predictions remain continuous even under long-range perception, avoiding common issues like fragmentation or topological adhesion.

#### 4.8.2. GENERALIZATION ON ARGOVERSE 2

Fig. 5 demonstrates generalization on the challenging Argoverse 2 dataset. Despite significant ground elevation variations (3D) and distinct sensor configurations, MapUQ accurately reconstructs unstructured road networks. This strongly evidences that uncertainty-driven feature learning effectively mitigates projection deviations caused by terrain

*Table 2.* **Ablation study of different components on NuScenes and Argoverse2 datasets.** "Unc-cls", "Unc-reg", and "Unc-seg" denote the Negative Sample Classifier, Progressive ROI Scale Adapter, and Multi-task Semantic Uncertainty Head, respectively.

| Dataset | Model | Components | | | AP (%) | | | | FPS |
|---|---|---|---|---|---|---|---|---|---|
| | | Unc-cls | Unc-reg | Unc-seg | Ped. | Div. | Bou. | Mean | |
| NuScenes | Baseline | | | | 59.8 | 62.4 | 62.4 | 61.5 | 14.1 |
| | MapUQ_1 | ✓ | | | 60.2 | 63.5 | 62.7 | 62.1 | 14.1 |
| | MapUQ_2 | | ✓ | | 60.8 | 62.5 | 62.6 | 62.0 | 14.0 |
| | MapUQ_3 | | | ✓ | 60.1 | 62.8 | 62.6 | 61.8 | 14.1 |
| | MapUQ_4 | ✓ | ✓ | | 61.0 | 63.0 | 62.8 | 62.3 | 13.9 |
| | MapUQ_5 | ✓ | | ✓ | 61.9 | 62.9 | 63.0 | 62.6 | 14.0 |
| | MapUQ_6 | | ✓ | ✓ | 61.5 | 63.1 | 63.1 | 62.6 | 13.9 |
| | **MapUQ_7** | ✓ | ✓ | ✓ | **62.3** | **63.2** | **63.6** | **63.0** | 13.9 |

*Table 3.* **Runtime breakdown analysis on NuScenes dataset.** The "Output Head" includes our proposed uncertainty modules, showing negligible computational cost.

| Dataset | Component | Runtime (ms) | Proportion |
|---|---|---|---|
| NuScenes | Map Encoder | 54.0 | 75.10% |
| | Map Decoder | 17.1 | 23.78% |
| | Output Head | 0.8 | 1.12% |
| | **Total** | **71.9** | **100.00%** |

*Table 4.* Quantitative evaluation of uncertainty estimation quality and effectiveness. **MapUQ** demonstrates superior calibration (lower ECE) and failure detection capability (lower AUSE).

| Method | Calibration | | Uncertainty Effectiveness | | |
|---|---|---|---|---|---|
| | ECE↓ | NLL↓ | AUSE↓ | AUROC↑ | FP Rate↓ |
| PivotNet | 14.2 | 3.37 | 23.4 | 85.8 | 16.2 |
| MapTR | 13.9 | 3.08 | 24.1 | 86.4 | 16.3 |
| MapTRv2 | 13.1 | 2.89 | 22.7 | 87.6 | 15.7 |
| **MapUQ (Ours)** | **9.3** | **1.96** | **19.9** | **91.8** | **8.4** |

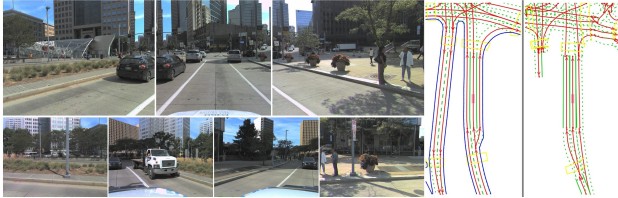

*Figure 5.* Visualization example in Argoverse 2 dataset.

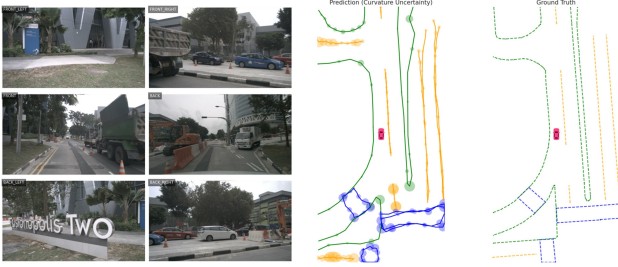

*Figure 6.* Visualisation results after adding uncertainty in NuScenes dataset.

undulations, showcasing robust cross-domain adaptability.

### 4.8.3. UNCERTAINTY VISUALIZATION

Fig. 6 visualizes the spatial distribution of predicted uncertainty, where bubble size indicates variance magnitude. High-uncertainty regions correlate strongly with physical blind spots, concentrating primarily on occlusions by vehicles or field-of-view edges. This confirms that the MSUH and NSC modules acutely capture aleatoric uncertainty, endowing the model with interpretability and providing reliable safety boundaries for downstream planning.

## 5. Conclusion

This paper focuses on autonomous driving safety in framework design and experimental quantification, aiming to address the challenges of constructing complex real-world scenarios. We propose MapUQ, a real-time BEV mapping framework utilizing MSUH, PRSA, and NSC for uncertainty quantification. Experiments on nuScenes and Argoverse 2 demonstrate that MapUQ outperforms baselines in NLL and ECE calibration metrics and surpasses state-of-the-art methods in AP, achieving a 1.5% average improvement.

## Impact Statement

We explore, for the first time comprehensively, the potential of uncertainty quantification in end-to-end frameworks, providing guidance for other researchers to further investigate UQ methodologies and their applications.

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

## A. Extended Experimental Results on Argoverse 2 Dataset

*Table 5.* **Comparison with state-of-the-art methods on the Argoverse 2 val set.** "Map dim" indicates the dimensionality of the vectorized map. **MapUQ** achieves superior performance in both 2D and 3D tasks compared to the baseline MapTRv2.

| Method | Map dim | Backbone | AP (%) | | | | FPS |
|---|---|---|---|---|---|---|---|
| | | | ped. | div. | bou. | mean | |
| PivotNet | 2 | R50 | 62.1 | 68.9 | 65.4 | 65.5 | 9.6 |
| MapTR | 2 | R50 | 53.9 | 62.3 | 57.8 | 58.0 | 14.1 |
| VectorMapNet | 2 | R50 | 38.3 | 36.1 | 39.2 | 37.9 | 3.1 |
| | 3 | R50 | 36.5 | 35.0 | 36.2 | 35.8 | 3.2 |
| MapVR | 2 | R50 | 54.6 | 60.0 | 58.0 | 57.5 | 14.2 |
| MapTRv2 | 2 | R50 | 62.9 | 72.1 | 67.1 | 67.4 | 12.1 |
| | 3 | R50 | 60.7 | 68.9 | 64.5 | 64.7 | 12.0 |
| **MapUQ (Ours)** | 2 | R50 | **63.7** | **72.9** | **68.0** | **68.2** | 11.9 |
| | 3 | R50 | **61.9** | **69.7** | **65.8** | **65.8** | 11.8 |

Table. 5 presents a comprehensive quantitative comparison of MapUQ against existing state-of-the-art methods on the Argoverse 2 validation set. This dataset is characterized by complex road topologies and substantial ground elevation variations, offering a rigorous testbed for geometric perception. MapUQ establishes new state-of-the-art performance in both 2D and 3D mapping tasks. A critical observation is that the performance gain in the 3D task (+1.1% mAP over the MapTRv2 baseline) is notably higher than that in the 2D task (+0.8%). This phenomenon strongly corroborates the physical interpretation that our uncertainty-aware mechanisms, particularly the PRSA module, effectively mitigate geometric projection errors induced by terrain undulations (Z-axis variations), which are more prevalent in Argoverse 2 than in nuScenes. Furthermore, in the "Pedestrian Crossing" category, known for its irregular shapes and blurred boundaries, MapUQ achieves remarkable APs of 63.7% (2D) and 61.9% (3D), significantly outperforming the baseline and verifying the efficacy of the MSUH module in handling semantic ambiguity.

To further dissect the source of these improvements, we conducted a detailed component-wise ablation study on the Argoverse 2 dataset, as shown in Table. 6. Starting from the MapTRv2-R50 baseline (64.7% mAP), the individual incorporation of the Unc-cls, Unc-reg, and Unc-seg modules yields incremental improvements of 0.2%, 0.3%, and 0.4%, respectively. These results indicate that each uncertainty head addresses a distinct type of perception error—semantic classification, geometric regression, and feature representation—exhibiting clear orthogonality without conflict. When all three modules are integrated (MapUQ_7), the model achieves a synergistic peak performance of 65.8% mAP. Crucially, this 1.1% total improvement is attained with a negligible reduction in inference speed (dropping only from 12.0 FPS to 11.8 FPS). This efficiency analysis demonstrates that MapUQ provides a highly practical solution for high-precision mapping in

*Table 6.* **Ablation study of different components on NuScenes and Argoverse2 datasets.** "Unc-cls", "Unc-reg", and "Unc-seg" denote the Negative Sample Classifier, Progressive ROI Scale Adapter, and Multi-task Semantic Uncertainty Head, respectively.

| Dataset | Model | Components | | | AP (%) | | | | FPS |
|---------|-------|-----------|---|---|--------|---|---|---|-----|
| | | Unc-cls | Unc-reg | Unc-seg | Ped. | Div. | Bou. | Mean | |
| Argoverse2 | Baseline | | | | 60.7 | 68.9 | 64.5 | 64.7 | 12.0 |
| | MapUQ_1 | ✓ | | | 60.9 | 69.0 | 64.9 | 64.9 | 12.0 |
| | MapUQ_2 | | ✓ | | 61.1 | 69.3 | 64.7 | 65.0 | 11.9 |
| | MapUQ_3 | | | ✓ | 61.3 | 69.0 | 65.0 | 65.1 | 12.0 |
| | MapUQ_4 | ✓ | ✓ | | 61.5 | 69.4 | 65.6 | 65.5 | 11.9 |
| | MapUQ_5 | ✓ | | ✓ | 61.4 | 69.5 | 65.3 | 65.4 | 11.8 |
| | MapUQ_6 | | ✓ | ✓ | 61.6 | 69.6 | 65.3 | 65.5 | 11.9 |
| | **MapUQ_7** | ✓ | ✓ | ✓ | **61.9** | **69.7** | **65.7** | **65.8** | 11.8 |

*Table 7.* **Ablation study on negative sample mining hyperparameters.** We analyze the impact of Dropout rate, the number of negative samples, and the discrimination threshold on model performance. The best result is highlighted in **bold**.

| Dropout rate | Number of negative samples | Threshold for distinguishing samples | | | | |
| --- | --- | --- | --- | --- | --- | --- |
| | | 0.3 | 0.4 | 0.5 | 0.6 | 0.7 |
| | 1 | 55.7 | 58.0 | 61.8 | 60.5 | 59.6 |
| 0.05 | 2 | 59.2 | 61.4 | 61.9 | 62.1 | 60.8 |
| | 3 | 54.0 | 57.1 | 58.7 | 57.5 | 56.1 |
| | 1 | 56.2 | 58.3 | 62.2 | 61.1 | 60.2 |
| 0.1 | 2 | 59.8 | 62.1 | **63.0** | 62.7 | 61.3 |
| | 3 | 54.3 | 57.6 | 59.5 | 58.1 | 56.5 |
| | 1 | 55.3 | 57.6 | 61.3 | 60.1 | 59.2 |
| 0.15 | 2 | 58.8 | 61.1 | 61.3 | 61.7 | 60.5 |
| | 3 | 53.8 | 56.7 | 58.2 | 57.1 | 55.9 |

complex, non-flat urban environments, successfully balancing robust uncertainty quantification with real-time requirements.

## B. Hyperparameter Sensitivity Analysis for Hard Negative Mining

To ascertain the optimal configuration for the uncertainty-guided hard negative mining strategy within the NSC module, we performed a comprehensive grid search across three pivotal hyperparameters: the MC-Dropout rate ($p$), the number of negative samples per positive query ($k$), and the discrimination threshold ($\delta$), as detailed in Table. 7. The experimental results regarding the dropout rate indicate that the model attains peak performance (63.0% mAP) at $p = 0.1$. A lower rate (0.05) appears to provide insufficient stochasticity, failing to adequately capture the epistemic uncertainty required for effective mining. Conversely, an excessively high rate (0.15) introduces detrimental noise that disrupts the semantic consistency of the features, resulting in a general performance degradation across all threshold settings.

Furthermore, the selection of the negative sample count exhibits a distinct "inverted U-shaped" trend, with optimal efficacy observed at $k = 2$. While introducing negative samples enhances discriminative capability, increasing the count to $k = 3$ precipitates a significant performance decline (e.g., dropping from 63.0% to 59.5% under the $0.1/0.5$ setting). This suggests that an excessive number of hard negatives may dominate the loss function, potentially skewing the optimization trajectory or introducing unreliable false negatives. Regarding the discrimination threshold, empirical results identify $\delta = 0.5$ as the ideal decision boundary that strikes the best balance between sample purity and recall. Thresholds that are either too strict ($> 0.6$) or too loose ($< 0.4$) lead to sub-optimal outcomes, validating the adoption of the $p = 0.1$, $k = 2$, and $\delta = 0.5$ configuration for our primary experiments.

## C. Hyperparameter Sensitivity Analysis for Progressive ROI Scale Adapter

*Table 8.* **Performance comparison under different hyperparameters.** We analyze the impact of scaling intensity coefficient ($\alpha$), temperature coefficient ($\tau$), and ROI scale limits on the PRSA module. The best performance is highlighted in **bold**.

| Scaling intensity coefficient ($\alpha$) | Temperature coefficient ($\tau$) | Upper and lower limits of scale ($[S_{min}, S_{max}]$) | | | | | |
|---|---|---|---|---|---|---|---|
| | | 0.5-1.2 | 0.5-1.5 | 0.7-1.2 | 0.7-1.5 | 0.9-1.5 | 0.9-1.7 |
| | 1.2 | 56.8 | 58.3 | 60.9 | 60.9 | 60.7 | 59.3 |
| 0.05 | 1.5 | 59.7 | 61.1 | 61.4 | 61.9 | 62.0 | 61.2 |
| | 1.8 | 56.8 | 57.8 | 58.1 | 58.3 | 57.8 | 55.4 |
| | 1.2 | 56.4 | 58.4 | 62.4 | 61.7 | 60.1 | 60.0 |
| 0.075 | 1.5 | 59.5 | 62.1 | 62.5 | **63.0** | 60.2 | 59.1 |
| | 1.8 | 55.8 | 58.1 | 59.5 | 59.5 | 58.1 | 56.5 |
| | 1.2 | 55.1 | 58.5 | 60.5 | 61.4 | 60.1 | 59.2 |
| 0.1 | 1.5 | 57.8 | 60.2 | 61.7 | 61.7 | 60.9 | 59.3 |
| | 1.8 | 54.4 | 56.3 | 58.9 | 59.1 | 57.3 | 55.9 |

Table. 8 presents a comprehensive performance evaluation of the PRSA module across varying configurations of the scaling intensity coefficient ($\alpha$), temperature coefficient ($\tau$), and ROI scale limits ($[S_{min}, S_{max}]$). Our empirical findings indicate that the adaptive adjustment of the Region of Interest requires precise constraints to balance responsiveness with stability. Specifically, the scaling intensity coefficient $\alpha$ exhibits a distinct "inverted U-shaped" trend regarding model performance. An excessively small value (0.05) results in insufficient adaptation, failing to adequately expand the receptive field for high-curvature geometries. Conversely, an overly large value (0.1) induces aggressive scale fluctuations, introducing detrimental noise and potential feature drift. Consequently, the configuration of $\alpha = 0.075$ combined with a temperature coefficient of $\tau = 1.5$ emerges as the optimal setting, achieving the highest baseline compatibility.

Furthermore, the boundaries of the ROI scale range prove critical for effective feature extraction. The experimental results demonstrate that the scale should be bounded within the interval $[0.7, 1.5]$. An overly narrow lower bound (0.5) causes excessive shrinkage of the receptive field in low-uncertainty regions, leading to the loss of local context essential for continuous lane reconstruction. On the other hand, an overly wide upper bound (1.7) extends the sampling range into non-road regions, thereby introducing irrelevant background noise that interferes with feature focusing. Ultimately, with the scale range constrained to $[0.7, 1.5]$, the model achieves optimal capture of geometric features, yielding a peak mAP of 63.0%, validating that a bounded adaptive strategy significantly outperforms static or unconstrained approaches.

# D. Additional Qualitative Visualization Results

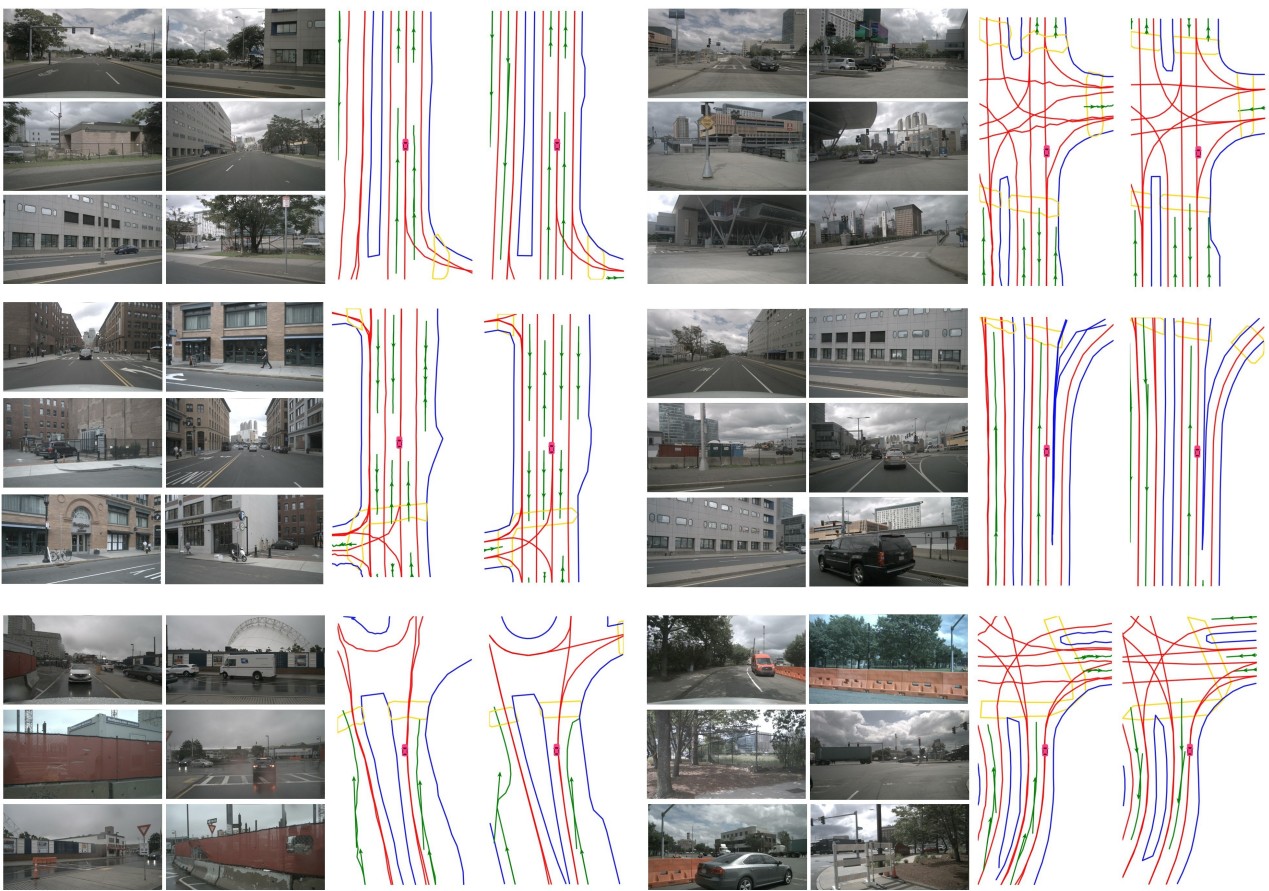

*Figure 7.* Additional visualisation results on the NuScenes dataset.

The MapUQ framework demonstrates exceptional topological stability on the nuScenes dataset, effectively handling a broad spectrum of driving scenarios ranging from urban canyons to intricate intersections (Figure. 7). Even in environments characterized by severe visual obstructions—such as construction barriers and heavy shadows cast by urban structures—the model successfully reconstructs continuous lane dividers and boundaries. This robust performance under environmental noise corroborates the effectiveness of the Multi-task Semantic Uncertainty Head (MSUH) in constructing noise-resilient feature priors, ensuring that the vectorization process remains undisturbed by transient visual artifacts. Furthermore, in high-density intersection scenarios involving multiple merging and diverging lanes, the model maintains precise geometric alignment with the ground truth, validating the capability of the Progressive ROI Scale Adapter (PRSA) to handle complex topologies.

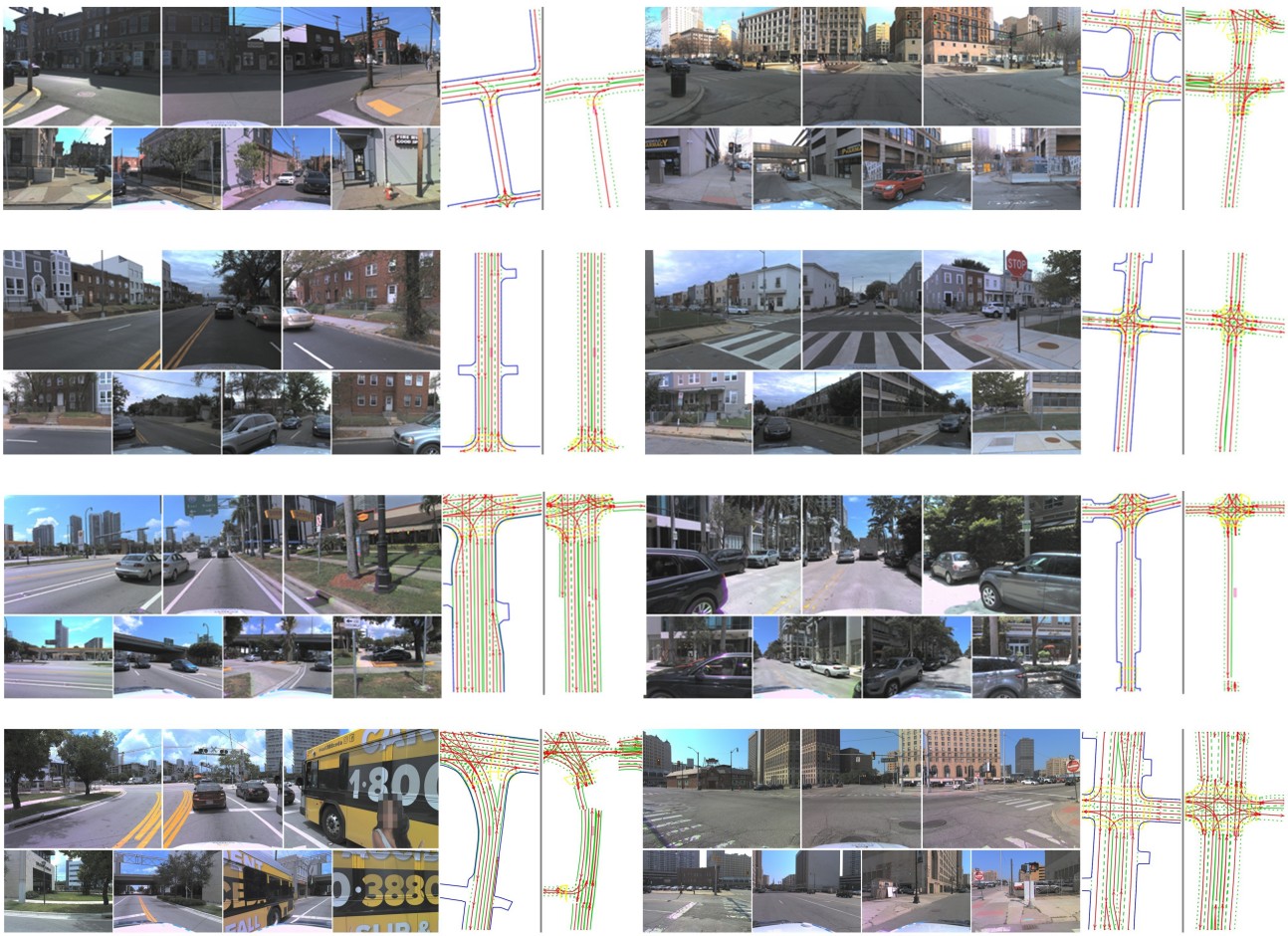

*Figure 8.* Additional visualisation results on the Argoverse 2 dataset.

The qualitative evaluation on the Argoverse 2 dataset highlights the exceptional generalization capability of MapUQ in handling substantial domain shifts and sensor discrepancies (Figure 8). Unlike the structured layouts of nuScenes, Argoverse 2 introduces unique difficulties, including high-resolution panoramic inputs, varying camera intrinsics, and topologically complex road networks. Despite these obstacles, MapUQ accurately reconstructs the road geometry in challenging scenarios without requiring extensive retraining or calibration.

For instance, as shown in the bottom-left example, the model successfully parses a highly intricate multi-lane intersection characterized by wide-angle roadways and merging ramps. In this scenario, where traditional fixed-scale reception often leads to broken lines, MapUQ maintains topological integrity without fragmentation. This success is largely attributed to the Progressive ROI Scale Adapter (PRSA), which dynamically expands the receptive field to capture long-range dependencies in these expansive curved lanes.

Similarly, in the top-right example, which depicts a dense urban canyon environment with potential GPS drift and strong shadows cast by high-rise buildings, the predicted vectorized lines adhere tightly to the ground truth. Here, the Multi-task Semantic Uncertainty Head (MSUH) plays a critical role by identifying and filtering out aleatoric noise caused by illumination changes, preventing the model from misinterpreting shadow edges as road boundaries. By effectively handling diverse road textures and sensor configurations across different cities, MapUQ proves that its quantitative SOTA performance translates into reliable, high-fidelity mapping in real-world deployments.

# E. Visualization and Interpretability of Spatial Uncertainty

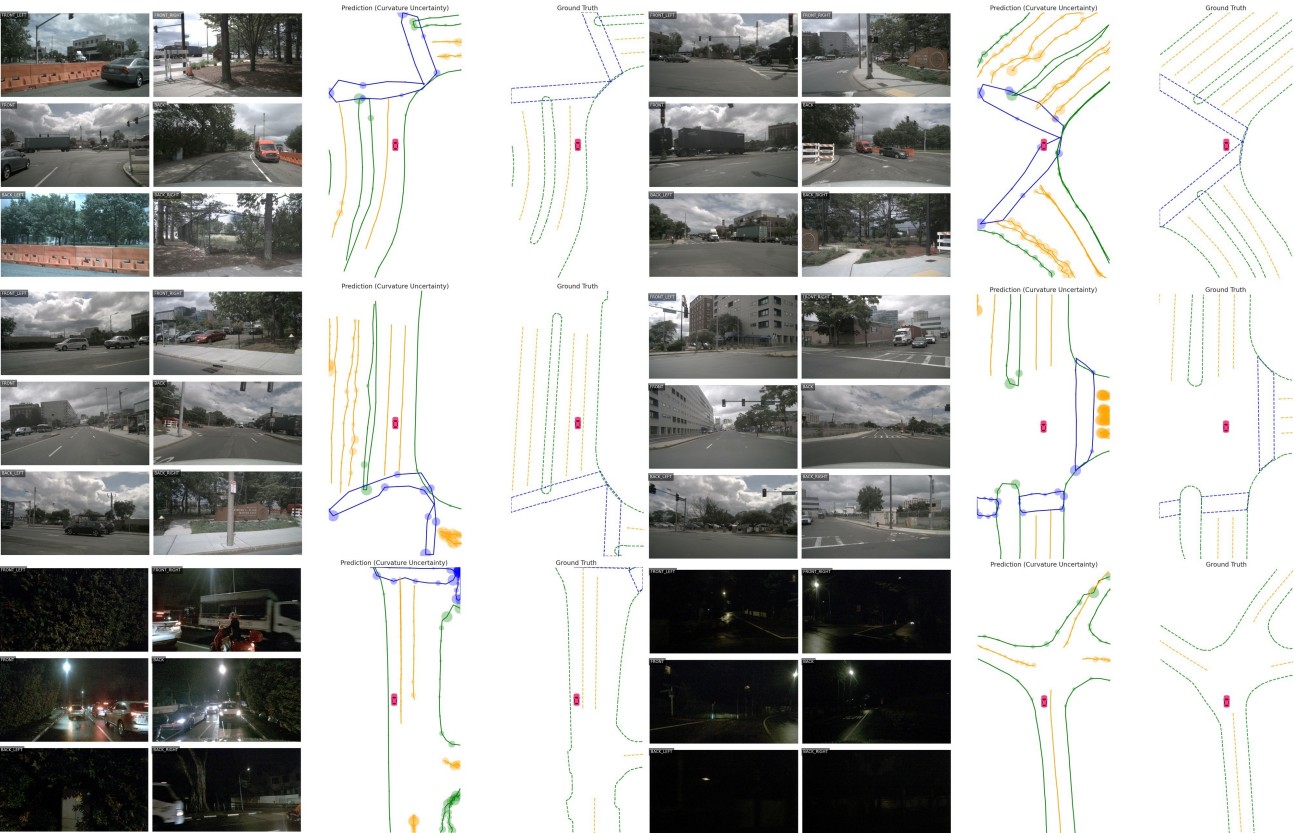

*Figure 9.* Extra examples of uncertainty visualisation on NuScenes dataset.

The interpretability of the MapUQ framework is further elucidated through an extended series of uncertainty visualizations on the nuScenes dataset (Figure. 9). In these visualizations, the spatial distribution of predictive variance—represented by the size of the bubbles along the vectorized lines—correlates strongly with environmental complexity and sensor limitations. Specifically, in the daytime scenarios presented in the top and middle rows, regions exhibiting high uncertainty are predominantly located in areas with severe visual occlusion, such as behind construction barriers, dense vegetation, or at the distal limits of the perceptual range. This precise localization of uncertainty confirms that the Multi-task Semantic Uncertainty Head (MSUH) effectively captures the aleatoric noise inherent in partial observability, preventing the model from making over-confident predictions in blind spots where geometric evidence is scarce.

Furthermore, the robustness of the uncertainty estimation mechanism is rigorously tested under challenging low-light conditions, as exemplified by the night scene in the bottom row. In such environments, where visual features are significantly degraded by low illumination, uneven street lighting, and sensor noise (low Signal-to-Noise Ratio), standard deterministic models are prone to hallucinating lane structures or missing them entirely. MapUQ, however, generates vectorized outputs that maintain topological consistency while appropriately assigning higher uncertainty scores to regions with ambiguous lane markings or glare interference. By explicitly quantifying the confidence degradation caused by poor lighting, the Negative Sample Classifier (NSC) ensures that the system remains "aware" of the reduced perception quality. This capability transforms the mapping output into a safety-critical risk map, enabling downstream planning modules to adopt more conservative strategies (e.g., increasing safety margins) when navigating through dark or visually noisy environments, rather than blindly trusting potentially flawed predictions.

