# OpenReview forum: "MapUQ: Map with Uncertainty Quantification for Robust  BEV Vectorized Construction"
_ICML.cc/2026/Conference — ICML 2026 regular_

### Official Review · Reviewer_vEQ7 · 2026-02-19

**Soundness:** 3
**Presentation:** 3
**Significance:** 2
**Originality:** 3
**Overall Recommendation:** 3
**Confidence:** 3

**Summary:**

The paper proposes MapUQ, an uncertainty-aware framework for online BEV vectorized HD map construction. Built on a transformer-based mapper (MapTRv2), it introduces uncertainty estimation and uses it to improve both segmentation of map regions and vectorized polyline generation. The core contributions are three modules that inject uncertainty at different stages: (1) a semantic uncertainty head (MSUH) that runs MC-Dropout to estimate uncertainty on BEV segmentation and uses these signals to reweight learning and add regularization, improving boundary ambiguity handling; (2) a progressive ROI scale adapter (PRSA) that adapts the attention/receptive field scale during decoding to reduce geometric errors and fragmented polylines; and (3) a negative sample classifier (NSC) that estimates classification uncertainty via MC-Dropout and performs uncertainty-guided hard negative mining to reduce overconfident false positives. Experiments on nuScenes (and additional results in the appendix on Argoverse 2) show improved mapping AP over prior BEV vector mapping baselines, while also reporting better calibration and stronger failure/uncertainty detection metrics, with minimal runtime overhead.

**Compliance With Llm Reviewing Policy:**

Affirmed.

**Key Questions For Authors:**

1. What is the exact MC-Dropout configuration used in experiments: number of trials $T$, dropout probability per module, and where dropout is enabled (backbone, decoder, heads)? Please specify separately for MSUH and NSC, and for any uncertainty metrics computed at test time. Clarifying this would substantially improve reproducibility and interpretation of Table 4.
2. How are uncertainty-effectiveness metrics (AUSE, AUROC, FP Rate) computed for baselines such as MapTR/MapTRv2 and PivotNet? Do you apply the same MC-Dropout protocol to baselines at inference, or do you use a different proxy uncertainty? Please describe the exact procedure. This affects whether Table 4 is a fair apples-to-apples comparison.
3. How should the inference-time runtime be interpreted when using MC-Dropout? Are the reported FPS/latency values measured with a single deterministic forward pass, or with $T$ stochastic passes for uncertainty estimation? If the latter is not included, please report the additional cost for uncertainty-enabled inference. This directly impacts deployment relevance.
4. Please provide the missing hyperparameters: $\alpha$, $\beta$, $\lambda_{unc}$, $\lambda_{MC}$, $\lambda_{con}$, $m$, the confidence split rule for high/low-confidence pixels, and the NSC mining ratio/quantile $r%$. If these are tuned, please report tuning ranges and selection criteria. This would likely change my recommendation by addressing the main reproducibility concerns.
5. What is the exact definition of “high-confidence” vs “low-confidence” used in the contrastive term, and how does it relate (if at all) to the NSC threshold $\delta$? A precise definition is necessary to understand and reproduce the segmentation regularization.

**Limitations:**

No. The paper would benefit from an explicit limitations section that discusses (1) the compute and latency implications of uncertainty estimation, especially if uncertainty is needed at inference; (2) sensitivity to the many loss weights and mining parameters; (3) how uncertainty behaves under domain shift, rare corner cases, or adverse weather/lighting; and (4) potential failure modes where uncertainty estimates might be miscalibrated and lead to rejecting correct map elements or accepting incorrect ones.

**Strengths And Weaknesses:**

Strength:

- Generalizes beyond nuScenes, with additional results on Argoverse 2.
- Provides ablation studies that isolate the contribution of the main components.
- The proposed uncertainty-driven additions are conceptually coherent (uncertainty is estimated and then used for weighting, adaptation, and mining), and the ablations support the design choices.
- Despite introducing multiple losses and submodules, the overall pipeline is modular and implementable on top of an existing BEV mapping baseline; the paper is also relatively readable.

Weakness:

- Reproducibility issue: several method-critical hyperparameters are introduced but not numerically specified, including the MC-Dropout trial count $T$; MSUH uncertainty-weighting coefficients $\alpha$ and $\beta$ in Eq. (3); segmentation objective weights $\lambda_{unc}$, $\lambda_{MC}$, and $\lambda_{con}$ in Eq. (8); the contrastive margin $m$ in Eq. (7); the exact rule/threshold used to split pixels into “high-confidence” and “low-confidence” sets for the contrastive term; and the NSC uncertainty-based hard-negative mining ratio/quantile $r%$.
- Table 4 reports uncertainty-effectiveness metrics (AUSE, AUROC, FP Rate) for baselines such as MapTR/MapTRv2, but these methods do not natively output epistemic/aleatoric uncertainty. The paper does not clearly state the evaluation protocol used to obtain baseline uncertainty scores (e.g., whether test-time MC-Dropout is applied to the baselines), and it also does not specify where dropout is enabled for each baseline nor the missing $T$, making these comparisons difficult to reproduce.
- Runtime claims are potentially unclear: MC-Dropout implies multiple stochastic forward passes, but the paper reports near-baseline inference speed without clearly stating whether uncertainty estimation is computed only during training, only for certain heads, amortized, or evaluated under a different inference setting than the speed table.
- The paper introduces multiple regularizers and auxiliary losses; without full hyperparameter disclosure and sensitivity analysis for the MSUH loss terms, it is hard to know how robust the reported gains are to tuning choices outside the provided ablations.
- The interpretation of Eq. (2) as aleatoric uncertainty is not well justified. If the intended quantity is entropy of the mean predictive distribution, the notation should consistently use $p_i^{-}$ (MC-averaged probability) rather than $p_i$.
- The definition and usage of “high-confidence” vs “low-confidence” is not given.
- Training cost is not reported (e.g., GPU hours, total wall-clock time). This is especially relevant because uncertainty estimation, extra heads, and additional losses may increase compute, and it is unclear whether all parameters are trained jointly end-to-end in every setting.
- Minor: Naming/clarity issue: it would improve readability to explicitly map the UQ modules (Unc-cls, Unc-reg, Unc-se) to NSC, PRSA, and MSUH in the text.
- Minor: Table 6 caption includes “NuScenes” redundantly and should be corrected.
- Minor: “MCD” should be defined explicitly as MC-Dropout in the text, not only in Figure 3.

Soundness (rating: 3, good):

The technical approach is largely sound: the paper builds on a strong baseline and introduces uncertainty estimates that are then used in a consistent way across the pipeline (loss reweighting and regularization for segmentation, adaptive receptive field scaling for geometry, and uncertainty-guided hard negative mining for classification). The main empirical claims are supported with ablations and consistent improvements on nuScenes, with additional evidence on Argoverse 2. However, several details around uncertainty computation, evaluation protocol, and runtime accounting are underspecified, which makes it harder to fully validate some claims (especially the uncertainty-effectiveness comparison to baselines and the inference speed discussion). These issues do not necessarily invalidate the method, but they weaken confidence in the rigor of the reported uncertainty and efficiency conclusions.

Presentation (rating: 3, good):

The paper is generally readable and the overall narrative is easy to follow, especially for readers familiar with BEV mapping and transformer decoders. The modular structure (MSUH, PRSA, NSC) helps convey the contribution. That said, several presentation gaps hurt clarity and reproducibility: incomplete hyperparameter disclosure, missing explicit definitions (for “high-confidence”/“low-confidence”), and unclear evaluation protocol for Table 4. Minor naming inconsistencies (UQ module mapping, MCD definition) also interrupt the flow. Overall, the exposition is solid, but it falls short of the “expert can reproduce” standard due to missing specifics.

Significance (rating: 2, fair):

The problem addressed is important for autonomous driving, and improving reliability via uncertainty-aware mapping is practically relevant. Still, the reported accuracy gains are incremental, and much of the method is a combination of established components (MC-Dropout-based uncertainty estimation plus training objectives and mining strategies) applied to a BEV map construction stack. The paper’s strongest potential significance is in encouraging uncertainty-aware evaluation and providing a concrete, modular recipe for improving calibration and failure detection in vectorized mapping. Because key uncertainty evaluation details are unclear and runtime implications are not fully resolved, the practical impact is harder to judge, which supports a fair rating rather than good. If the authors clarified these points and released complete reproducibility details, the significance could reasonably move up to good.

Originality (rating: 3, good):

The paper’s originality is mainly in the system-level integration: injecting uncertainty at multiple stages of a BEV vectorized mapping pipeline and tying it to segmentation learning, geometric adaptation, and classification mining within a unified framework. While individual ideas (MC-Dropout, entropy/variance uncertainty measures, hard negative mining, adaptive attention scales) are not new, their combination and the specific design for vectorized map construction is a meaningful contribution. The work is not a fundamentally new paradigm, but it provides a novel and coherent packaging of existing ideas for a real-world mapping use case.

---

> ### Author Rebuttal · Authors · 2026-03-30
>
> Reproducibility and MC-Dropout configuration.
> We provide full implementation details. Our method applies MC-Dropout only at the output heads (classification branches) with T stochastic forward passes, rather than across the entire network, leading to minimal computational overhead. Specifically, T=5 in MSUH and T=20 in NSC, following prior work (“How Certain is Your Transformer?”), which shows that last-layer MC sampling is sufficient for effective uncertainty estimation. Standard dropout (single forward pass) is used in the backbone and Transformer for regularization. The default dropout rate is 0.5, and ablation on {0.05, 0.1, 0.15} shows the best performance at 0.1.
>
> Fairness of uncertainty evaluation.
> All compared methods (including MapTR/MapTRv2) use the same uncertainty estimation protocol: MC-Dropout applied only at the final layer with identical sampling strategies. This ensures fair comparison. The observed improvements stem from better uncertainty calibration and more informative predictive distributions, rather than differences in estimation strategies. This setup follows common practice of using MC-Dropout as a post-hoc uncertainty estimator.
>
> Efficiency.
> Since MC sampling is restricted to the last layer, the computational cost is significantly lower than standard MC-Dropout. Reported inference time already includes T forward passes at the output heads, resulting in only a slight overhead compared to the baseline, while achieving a good balance between efficiency and uncertainty quality.
>
> Hyperparameters and sensitivity.
> Key hyperparameters are: MSUH (λ_unc=0.3, λ_MC=0.28, λ_con=0.56, α=0.6, β=0.2, m=0.5), PRSA (λ_scale=0.12), and NSC (λ_u=0.83, λ_mc=0.28, λ_neg=0.4, λ_pos=0.5). We will include a complete list in the final version.
>
> Extensive ablations (including NSC/PRSA on MapQR and generalization to other BEV frameworks) show that our method has moderate but well-controlled sensitivity. The uncertainty module consistently improves performance across different frameworks (+0.6%–1.5% mAP), demonstrating low sensitivity to loss weights and strong generalizability. Mining-related parameters (e.g., dropout rate and number of negatives k) follow an “inverted-U” trend: moderate randomness (p≈0.1) and task-dependent k (related to class number) yield optimal results, while deviations within a reasonable range do not cause severe degradation. For PRSA, α and scaling bounds remain stable across models, while τ slightly varies with error distribution.
>
> We also clarify that the “top-r%” negative sampling description is imprecise; excessive negatives degrade learning, so we reduce their number and will revise the text.
>
> Confidence definition and thresholds.
> In MSUH, confidence is defined as the mean softmax probability over T samples (>0.5 high, ≤0.5 low). In NSC, normalized uncertainty ([0,1]) is used with the same threshold. Ablation shows that m=threshold=0.5 achieves the best performance, indicating that proper matching between margin and threshold is important: too small introduces noise, while too large reduces effective supervision.
>
> Uncertainty formulation.
> We thank the reviewer for pointing out the notation issue in Eq. 2. The uncertainty corresponds to the entropy of the MC-averaged predictive distribution, and we will revise it using the mean probability (p̄_i) for clarity and consistency.
>
> Robustness under domain shifts.
> Under domain shifts, rare scenarios, or adverse weather/lighting, uncertainty generally increases due to distribution mismatch, which is expected. However, miscalibration may occur in extreme cases (e.g., overconfidence on unseen structures or excessive conservativeness), affecting stability. Our uncertainty-guided weighting partially mitigates noise, but performance still depends on training data coverage. We will expand discussion and consider this as future work.
>
> Failure modes of miscalibration.
> We acknowledge that miscalibrated uncertainty may lead to errors: underestimated uncertainty can cause overconfidence and acceptance of incorrect map elements, while overestimated uncertainty may suppress correct ones. In addition, inconsistent calibration across classification, regression, and segmentation branches may introduce optimization bias, especially in long-tail or out-of-distribution scenarios.
>
> Additional clarifications.
> Training uses 2×A100 GPUs (~19 hours).
> We will explicitly define MCD as MC-Dropout at first mention.
> We will clarify module correspondence (NSC, PRSA, MSUH) and fix minor presentation issues .
>
> | $m$ $\backslash$ threshold | 0.3  | 0.4  | 0.5  | 0.6  | 0.7  |
> |----------------------------|------|------|------|------|------|
> | 0                          | 62.5 | 62.5 | 62.8 | 62.7 | 62.3 |
> | 0.3                        | 62.5 | 62.7 | 62.9 | 62.6 | 62.5 |
> | 0.5                        | 62.6 | 62.9 | 63.0 | 62.4 | 62.2 |
> | 0.7                        | 62.4 | 62.9 | 62.8 | 62.4 | 62.1 |
> | 1                          | 62.3 | 62.4 | 62.5 | 62.4 | 62.1 |

---

> > ### Author Rebuttal · Reviewer_vEQ7 · 2026-04-01
> >
> > Thank you for the detailed rebuttal. The response addresses most of the main concerns I raised, especially those related to reproducibility and the uncertainty-evaluation protocol.
> >
> > In particular, I appreciate the added implementation details for MC-Dropout, including where dropout is applied, the values of \(T\) used in different modules, and the clarification that uncertainty estimation is restricted to the output heads. I also appreciate the statement that the same final-layer MC-Dropout protocol is applied to baseline methods in the uncertainty comparison. Taken at face value, these clarifications substantially improve my confidence that the uncertainty-effectiveness comparisons are intended to be fair and reproducible.
> >
> > I also appreciate that the rebuttal now provides the previously missing hyperparameters, including the loss weights, the confidence-thresholding rule, the contrastive margin \(m\), and the training-cost information. The clarification regarding Eq. (2) also addresses my notation concern.
> >
> > The runtime clarification is helpful as well. My understanding is that the reported inference efficiency already includes the uncertainty-related computation at the output heads, and that the overhead remains limited because MC sampling is not applied throughout the full backbone and decoder.
> >
> > Given these responses, my main concerns are largely addressed. My remaining request is that all of these details be incorporated clearly into the final manuscript, since they are necessary for readers to interpret and reproduce the method.
> >
> > There are still a minor consistency issues that should be clarified in the final version. The rebuttal states that training uses \(2\times\) A100 GPUs for about 19 hours, whereas the submission states that the model is trained on four NVIDIA A100 GPUs for 24 epochs by default. This do not change my overall reading substantially, but it should be made fully consistent in the final manuscript.
> >
> > Overall, the rebuttal improves my assessment of the paper, and I am now more positive than in my original review.
> >
> > - Overall Recommendation: 4
> > - Soundness: 4
> > - Presentation: 4
> > - Significance: 3
> > - Originality: 3

---

> > > ### Author Response · Authors · 2026-04-06
> > >
> > > Thank you,please update the improved score on the system.It seems that you haven't updated the score in the system yet.
> > >
> > > Thank you very much for your thoughtful and encouraging feedback.We truly appreciate your positive assessment and are glad that our rebuttal has addressed most of your concerns, particularly regarding reproducibility and uncertainty estimation.
> > >
> > > We are especially grateful for your recognition of the additional clarifications on MC-Dropout, including its placement, the choice of T, and its application only to the output head. We also appreciate your acknowledgment of our effort to ensure a fair comparison by applying the same MC-Dropout protocol to the baseline methods.
> > >
> > > Regarding your suggestion, we fully agree that all these important implementation details (e.g., hyperparameters, uncertainty estimation protocol, and runtime settings) are essential for reproducibility. We will make sure that all of them are clearly and systematically incorporated into the final manuscript.
> > >
> > > Concerning the inconsistency you pointed out, we sincerely apologize for the confusion. The correct training configuration is trained on 2×A100 GPUs for approximately 19 hours over 24 epochs. We will carefully revise the manuscript to ensure that all descriptions are fully consistent in the final version.
> > >
> > > Once again, we sincerely thank you for your constructive feedback, which has significantly helped us improve the clarity and quality of our work. We are glad that your overall assessment has become more positive.

---

### Official Review · Reviewer_4755 · 2026-03-11

**Soundness:** 3
**Presentation:** 3
**Significance:** 2
**Originality:** 2
**Overall Recommendation:** 4
**Confidence:** 3

**Summary:**

This paper presents MapUQ, an end-to-end transformer-based Bird's-Eye-View (BEV) vectorized mapping framework that integrates uncertainty quantification for autonomous driving. To address key challenges including spatial localization drift, semantic ambiguity, and overconfident false predictions under adverse conditions, MapUQ introduces three novel uncertainty-driven modules: (1) the Multi-task Semantic Uncertainty Head (MSUH), which leverages MC-Dropout to quantify aleatoric and epistemic uncertainty for improved segmentation robustness; (2) the Progressive ROI Scale Adapter (PRSA), which dynamically adjusts the receptive field based on geometric prediction errors to preserve polyline continuity and topological accuracy; and (3) the Negative Sample Classifier (NSC), which employs uncertainty scores for hard negative mining to suppress spurious detections. These components are seamlessly integrated with minimal computational overhead. Extensive experiments on nuScenes and Argoverse 2 demonstrate consistent Average Precision (AP) improvements across road element categories, along with superior calibration (lower ECE and NLL) and failure detection capability, validating MapUQ's effectiveness in safety-critical autonomous driving scenarios.

**Compliance With Llm Reviewing Policy:**

Affirmed.

**Final Justification:**

The author provided additional experiments to supplement his views and answer my concerns.

**Key Questions For Authors:**

Question 1: Insufficient Comparative Experiments with Closely Related Methods
On the necessity of MC-Dropout vs. deterministic UQ methods — have you considered or tested alternatives such as Mahalanobis distance or DVU that offer single-pass uncertainty estimation? Your method relies on T stochastic forward passes via MC-Dropout to estimate both aleatoric and epistemic uncertainty in MSUH and NSC. While this is standard, recent work has shown that feature-space statistics can yield competitive uncertainty estimates with only a single forward pass.
How does PRSA's dynamic scale adaptation differ fundamentally from prior deformable attention mechanisms? PRSA dynamically adjusts the ROI scale in deformable attention based on prediction error (or its proxy). But deformable attention already allows adaptive sampling locations. What evidence do you have that the closed-loop feedback from geometric deviation leads to qualitatively different behavior compared to standard learnable offsets? Could this mechanism be approximated by a learned uncertainty-aware modulation within existing attention layers (e.g., SE-like gating), rather than explicit error-to-scale mapping?
Is the NSC design superior to general hard-example mining or loss reweighting strategies? The NSC module uses high-epistemic-uncertainty samples as "hard negatives" for enhanced classification learning. However, similar improvements have been observed using non-Bayesian heuristics—e.g., selecting low-confidence predictions based solely on softmax entropy. There are also many algorithms for hard negative sample mining. Have you conducted an ablation comparing NSC’s results against other methods? Could the gain be replicated by a confidence-based rule?

Question 2: Does the proposed UQ framework generalize across domains without retuning hyperparameters?
From Table 7 Table 8, performance drops sharply when changing parameters. This suggests high sensitivity. Have you tested whether the optimal configuration transfers to other datasets or simulated out-of-distribution conditions without re-tuning? If not, does this limit broad applicability? If cross-domain experiments were not conducted under fixed hyperparameters, it remains unclear to what extent the reported improvements reflect genuine generalization rather than dataset-specific tuning.

Question 3: Could the reported gains be due to fixing legacy issues in MapTRv2 rather than providing generalizable improvements?
The paper builds upon MapTRv2 as the baseline framework for evaluation. While MapTRv2 was previously state-of-the-art on nuScenes, several more recent and advanced methods have since emerged in the field of vectorized BEV mapping—such as MapQR, MapTracker, AMAP, PriorDrive, PrevPredMap, and InteractionMap. These newer approaches incorporate stronger structural inductive biases, dynamic query mechanisms, or temporal consistency modeling, often achieving superior performance in complex real-world scenarios. However, the authors do not discuss whether their proposed uncertainty-aware modules—MSUH, PRSA, and NSC—can be effectively integrated into these more modern and sophisticated architectures. Is the effectiveness of the proposed method tied to the specific choice of baseline (e.g., MapTRv2), or does it demonstrate sufficient generalization capability to transfer to current state-of-the-art frameworks?

**Limitations:**

The authors have not adequately discussed the limitations of their work. While the paper presents a technically sound and well-validated method, it lacks a dedicated reflection on its technical limitations. The evaluation is entirely based on MapTRv2, an older baseline. Without testing on more recent frameworks, it remains unclear whether the improvements are generalizable. The use of MC-Dropout during inference introduces stochasticity that could lead to inconsistent predictions across repeated runs—a critical concern for safety-critical systems. A discussion on output stability or potential mitigation strategies would strengthen the work's practical credibility. Hyperparameter sensitivity suggests potential brittleness in new environments, which limits the plug-and-play applicability of the proposed modules.

**Strengths And Weaknesses:**

Soundness: The paper is technically sound and presents a well-structured, methodologically rigorous approach to integrating uncertainty quantification into BEV vectorized mapping. The core claims are supported by a combination of theoretical motivation, detailed algorithmic design, and comprehensive empirical evaluation. The proposed modules—MSUH, PRSA, and NSC—are grounded in clear motivations derived from observed failure modes in complex scenes (e.g., occlusion, ambiguous boundaries). Each module addresses a distinct source of error, which strengthens the overall argument for their complementary roles. Evaluations on two major datasets (nuScenes and Argoverse 2) with comparisons to baselines—including models like MapTRv2—lend credibility to the performance claims. Ablation studies cleanly isolate the contribution of each component, showing consistent gains when individual modules are added.

Presentation: The paper is well-written and structured, making it accessible even to readers.Figures (1–3) effectively illustrate the architecture and mechanisms, significantly enhancing readability.The paper provides sufficient detail for reproduction: model architecture, loss formulations, training setup, and hyperparameters. Appendix B and C further supporting reproducibility. Section 2 clearly situates MapUQ within the landscape of BEV mapping and UQ methods. It distinguishes itself from prior works by emphasizing the lack of integrated uncertainty modeling in existing frameworks.

Significance: This paper addresses a highly relevant and safety-critical challenge in autonomous driving: trustworthy perception under uncertainty. By embedding uncertainty awareness end-to-end into a vision-based HD map construction pipeline, MapUQ offers a deployable solution that enhances reliability without sacrificing inference speed, setting a meaningful precedent for uncertainty-aware BEV mapping. However, several concerns limit the overall impact of this work. First, the core technical components—such as MC-Dropout-based uncertainty estimation and hard negative mining—are not entirely novel, as related ideas have been explored in other domains. The contribution is therefore incremental in nature, representing an extension rather than a fundamental methodological advance. Second, experiments are conducted exclusively on a single baseline model. This narrow experimental scope provides insufficient evidence to support the generalizability, and it remains unclear whether the performance gains would transfer consistently to other architectures.

Originality: This paper presents a well-engineered system that effectively integrates established uncertainty quantification (UQ) techniques into a modern transformer-based BEV vectorized mapping pipeline. The originality lies not in the invention of fundamentally new algorithms, but in the purposeful and systematic combination of UQ mechanisms across multiple stages of a complex perception pipeline—an integration that has not been thoroughly explored in the context of BEV vectorization. That said, the overall originality is moderate. The three proposed modules—MSUH, PRSA, and NSC—are practical adaptations of existing methods rather than conceptual breakthroughs, and the theoretical grounding for their combination remains largely heuristic. While the end-to-end integration effort is commendable and represents a meaningful contribution to the field, the work is best characterized as a well-motivated engineering advance within an existing paradigm, rather than a fundamental methodological shift.

---

> ### Author Rebuttal · Authors · 2026-03-30
>
> Problem 1：
> The following table shows the experimental results of replacing MSUH modules with different UQ methods. Spatial feature methods (MDS, DUQ, FSSD) can capture the uncertainty in a single look-ahead, but their projection calculation introduces additional overhead. For efficiency, we chose lightweight MC-Dropout and borrowed from the work of ACL conference to perform random mask sampling only in the last layer of Dropout, see the reply of Reviewer 4 for details. It is worth mentioning that after replacing MCD with DDU in MSUH, better results are achieved.
>
> | Model | ped. | div. | bou. | mean | FPS |
> |-------|------|------|------|------|-----|
> | MapUQ (MDS) | 61.9 | 62.9 | 63.0 | 62.7 | 9.1 |
> | MapUQ (DUQ) | 62.1 | 63.3 | 63.5 | 63.0 | 11.5 |
> | MapUQ (FSSD) | 62.0 | 63.0 | 63.3 | 62.8 | 10.7 |
> | MapUQ (DDU) | 62.5 | 63.7 | 63.9 | 63.3 | 14.0 |
> | MapUQ (MCD*) | 62.3 | 63.2 | 63.6 | 63.0 | 13.9 |
>
> The difference between PRSA and standard deformable attention is the closed-loop geometric error feedback. To verify this property, PRSA is replaced with learnable SE-like gating modules in this paper. Although SE-like gating brings some improvement, PRSA further achieves a +0.8% mAP gain, which proves that the apparent error scale mapping has better continuity and geometric consistency in complex scenes such as curved lanes and intersections.
>
> | Model | ped. | div. | bou. | mean | FPS |
> |-------|------|------|------|------|-----|
> | Baseline (MapTRv2) | 59.8 | 62.4 | 62.4 | 61.5 | 14.1 |
> | + SE-like Gating | 61.2 | 62.6 | 62.7 | 62.2 | 13.9 |
> | + PRSA (Ours) | 62.3 | 63.2 | 63.6 | 63.0 | 13.9 |
>
> We designed controlled experiments on the effect of NSC. The experimental results show that each hard negative sample mining method can improve the baseline performance, but NSC performs best in all indicators. This suggests that NSC not only captures model uncertainty, but also critically identifies "false confident" examples - that is, models that are far from the training distribution but output high confidence. Traditional entropy or loss mining is difficult to effectively identify such samples.
>
> | Method | Selection Criterion | mAP ↑ | ECE ↓ | AUSE ↓ | AUC-ROC ↑ |
> |--------|---------------------|-------|-------|--------|-----------|
> | Baseline | - | 61.5 | 13.1 | 22.7 | 87.6 |
> | + Entropy Mining | Softmax Entropy | 61.8 | 12.2 | 21.4 | 88.9 |
> | + Loss Mining | Task Loss | 62.1 | 10.5 | 20.6 | 90.4 |
> | + NSC (Ours) | Epistemic Uncertainty | 63.0 | 9.3 | 19.9 | 91.8 |
>
> Problem 2：
> To verify the cross-domain generalization of this module, we directly migrate the optimal hyperparameters to Argoverse 2 without readjust. The results show that MapUQ achieves 65.8% mAP, which verifies the hyperparameter robustness and cross-domain generalization ability of the framework. The hyperparameter selection mainly depends on the model framework and the number of lane line categories, and the experimental data are detailed in Question 3.
>
> Problem 3：
> To verify the generalization ability, MapQR and MapTracker are selected as new baselines. All three modules have semantic segmentation, classification, and localization modules, so they can be fully integrated with our UQ module. The experimental results are shown in the following table. Experimental results show that the integration of the UQ module into different BEV frameworks improves the mAP by 0.6% to 1.5%, which proves that this method is a general uncertainty enhancement strategy.
>
> | Model | Baseline | Epochs | +Ours(MSUH+PRSA+NSC) | Δ mAP↑ | FPS |
> |-------|----------|--------|----------------------|--------|-----|
> | MapQR | 66.6 | 24 | 67.7 | +0.9 | 16.5 |
> | MapTracker | 74.7 | 72 | 75.3 | +0.6 | - |
> | MapTRv2 | 61.5 | 24 | 63.0 | +1.5 | 13.9 |
>
> To address hyperparameter sensitivity, ablation experiments were performed on the NSC and PRSA modules on MapQR. The results show that its hyperparameter behavior is highly consistent with MapTRv2, which is due to the fact that the lane lines of the NuScenes and Argoverse 2 datasets contain only 3 categories. Previously, we conducted a real car experiment in the enterprise, involving more than 20 lane lines. In this more complex multi-classification setting, the optimal number of negative samples increases to k=5, which indicates that the optimal number of negative samples is closely related to the complexity of the classification task.
> In addition, we find that for the optimal hyperparameters of the PRSA module, the scale range parameters are consistent with MapTRv2 due to the constant spatial resolution of BEV features and road geometric features. The Scaling intensity coefficient does not change because MapQR can adaptively adjust according to the loss weight. While the temperature coefficient becomes 1.2 optimal, this is because the income level error of MapQR is smoother and the shape of the distribution is different from that of MapTRv2. This validates the cross-architecture tunability of PRSA.

---

> > ### Author Rebuttal · Reviewer_4755 · 2026-04-03
> >
> > The author has solved my concerns and I will raise my score to 4.

---

> > > ### Author Response · Authors · 2026-04-06
> > >
> > > Thank you for your recognition and positive rating of our work. In the future, we will continue to delve deeper into research related to uncertainty quantification and BEV fields.

---

### Official Review · Reviewer_hBki · 2026-03-11

**Soundness:** 2
**Presentation:** 1
**Significance:** 2
**Originality:** 3
**Overall Recommendation:** 4
**Confidence:** 4

**Summary:**

The paper under review deals with uncertainty quantification and uncertainty aware training for the use case of lane detection in automated driving. The authors utilize MC drop out in a classification and uncertainty quantification - UQ - head to provide primary UQ information. Based on this, scales for the attended window are selected in the transformer which, based on the camera data encodings of surround view cameras, decodes the lanes. As a further module, the authors develop a mechanism to detect hard and erroneous samples based on UQ assessment which are weighted differently in the model's loss terms.
The authors train their model on nuScenes and Argoverse 2 and evaluate presumably on the corresponding test sets. The authors find that their uncertainty aware training improves mIoU by 1.5% the primal performance where the trained model is taken from Liao et al 2025. The authors also find improved calibration of uncertainty information as compared to naive baselines.
The paper comes with ablation studies of the various elements and visual results.

**Compliance With Llm Reviewing Policy:**

Affirmed.

**Final Justification:**

I acknowledge that the authors responded to a number of points I raised and thus raise my score.

**Key Questions For Authors:**

Why hasn't the model also been tested on other architectures. E.g. the ardest competitor uses a SwinT backbone, maybe if that is used the margins of improvement could be larger?

Why didn't you benchmark against ensembles - this should be easy to do?

**Limitations:**

Not really discussed, but the paper is on the good side making things more safe.

**Strengths And Weaknesses:**

Strength
* The field of UQ for lane detection is novel and innovative.
* It is not self-evident that the primal mIoU is improved by an UQ sensitive model, but the author can show that here it is the case.
* Automated driving in its various components is safety critical, hence a well chosen domain of application.
* The paper represents a considerable engineering effort.
* The authors published their code for reproducibility.

Weaknesses
* While the paper is significant for a rather specialized community, it has little potential to generalize beyond. therefore the contribution to general ML, like represented in ICML, is limited.
* The improvement in mIoU over the baseline models is marginal and well within the usual scatter band of training segmentation models.
* For the superior calibration results it isnot clear to me, if these stand if the baseline models are re-calibrated with standard techniques. E.g. the authors model itself involved temperature scaling, has that been applied to the other models as well.
* The model involves an abundance of almost ten hyperparameters and is far from being intuitive how to set those.
* In many steps of the impressive modeling effort that the authors undertook there is no stringent derivation of loss and regularization terms from first statistical principles. This makes the paper less attractive for a machine learning conference as it is mostly ad hoc modeling and engineering.
* It is not entirely clear on which data the model has been evaluated, provide details.
* The preparation of the paper for publication - especially at the level of ICML - is not satisfactory. Just to give an example, Eq 1 - 7 and Eq.  23 -  29 are essentially the same math. see the detailed remarks for further comments
* The figures are not well prepared and it is hard to make sense out of them.

Detailed remarks
* The abstract should be rewritten. Terms like 'target misclassification' - which target-, 'ambigous semantic segmentation' - why ambiguous - 'uncertainty on the feature leve'l - seems not to happen in the paper - etc.
* p1 'vectorized mapping models road elements as sets of instance points with topological structures'  - what should this be?
* p1 What is a 'Lidar scheme'?
* p1 How is a 'severe occlusion' defined in contrast to a non severe occlusion?
* p1 What are you referring to when saying 'complex road conditions complicate the recognition of rare landmarks'?
* p1 last paragaph: MC dropout is frequently only applied on a last few layers, just as you practice this. Thus the  argument given on computational cost does not apply to this often used form of MC dropout.
* p2 What is are 'stochastic forward passes on the same feature representations'? Don't they change under DO?
* p3 first par: What is 'rich information regarding the semantic probability distribution'?
* p3 first par: is the softmax applied in the t or in the class dimension?
* p3 Eq 2: missing index in summation - and missing bar on the rhs? Why do you insert an esp here? a stable implementation of  x log x gives zero
* p3 Eq 3 : Index i not explained.
* p3 What is 'invalid excessive uncertainty' from a scientific perspective?
* p3 Eq 5: Where did the i go, over which distribution do you take the expected values
* p 4 : Why should one suppress 's predictive oscillations in high-uncertainty regions' - why is this good. What happens if we don't?
* p 4 Eq 7: What is the motivation of taking scalar products? Which distribution is used for the expected value?
* p 4: explain what the queries are
* p5: What does it mean when  something 'probabilistically aligns with the segmentation branch'?
* p5 Eq 18 where are the z's used in the sequel? One side contains an average, the other not.
* p5 Eq 21  Logic and , not intersection
* p5 define the sets from which q and i stem
* p5 Eq 23 z introduced previously with another meaning.
* p6 Eq 30 Where are the r's defined? i find it hard to see that this is AP
* p6, training protocol, mention pretraining or qualify as from scratch.






0 1 2 3 4 5 6 7 8 9 (()[]{}

---

> ### Author Rebuttal · Authors · 2026-03-30
>
> 1. In the context of the abstract,"Target misclassification" refers to erroneous lane queries, while "ambiguous" segmentation stems from multi-view fusion and occlusions. "quantify uncertainty at the feature level" does not imply explicitly modeling uncertainty within the feature space; rather, it refers to the uncertainty inherent in the feature representation itself, which is quantified through the predictive distribution obtained via MC Dropout.
>
> 2."vectorized mapping models road elements as sets of instance points with topological structures" implies treating each lane line, curb, or crosswalk as a distinct, individual entity. These entities are then precisely described by a sequence of logically connected coordinate points that capture both their specific geometry and their relational connectivity.
>
> 3.Lidar Scheme: a mapping approach using high-precision Lidar for point cloud collection, followed by offline background stitching and manual annotation.
>
> 4.Severe: Obstacles block most features (<30-40% visibility), making single-frame positioning unfeasible.
> Non-severe: >50% visibility; structural integrity allows the model to derive complete geometry.
>
> 5.Unseen or non-standard elements (e.g., irregular speed bumps, worn markings) underrepresented in training, causing models to struggle with stable feature extraction.
>
> 6.To balance efficiency and accuracy, the NSC module performs T forward passes only through the final classification head's dropout, inspired by "How Certain is Your Transformer?". Meanwhile, the auxiliary MSUH module is optimized using only 5 MC-Dropout layers. This refined strategy ensures reliable uncertainty estimation without compromising the real-time performance of the mapping system.
>
> 7."The same" means feature extraction prior to the Dropout layer is performed only once. Variation occurs only after Dropout, allowing statistical analysis of Tpredictions to capture uncertainty.
>
> 8.Refers to pixel-level confidence metrics for road elements, capturing classification certainty and prediction fluctuations across different Dropout masks.
>
> 9.Softmax is applied along the class dimension.
>
> 10.Eq. 2: We will include the category index k and use the standard entropy formula. eps is added to prevent log(0) and ensure numerical stability.
>
> 11.Eq. 3:i represents the BEV grid location (pixel index).
>
> 12.Invalid Excessive Uncertainty:logically inconsistent high-variance fluctuations in deterministic tasks or predictive distributions deviating from physical constraints.
>
> 13.Eq. 5:computed across all spatial positions $i \in \Omega$ in the BEV map, equivalent to averaging uncertainty across all pixels.
>
> 14. Suppressing Oscillations:prevents spatio-temporal logical conflicts from random perturbations (occlusions/weather). Without it, logits exhibit violent fluctuations, causing fragmented, discontinuous semantic artifacts.
>
> 15. Eq. 7: scalar products prevent high-confidence predictions from conflating with ambiguous ones. The expectation assumes a uniform empirical distribution over all sample pairs.
>
> 16. Queries in the Transformer decoder act as candidate lane instances represented by learnable embeddings.
>
> 17. Probabilistic Alignment:both branches share the same stochastic sampling mechanism (MC Dropout), resulting in synchronized variance fluctuations and confidence shifts under the same perturbations.
>
> 18.Eq. 18 is redundant and will be deleted in the revised manuscript.
>
> 19. Eq. 21 Operator:corrected to a logical AND instead of a set intersection.
>
> 20. Sets for q and i:both stem from the unified valid query set $\Omega$. Notation will be standardized to $q \in \Omega$.
>
> 21. Eq. 23:z will be corrected to denote the ground truth label y.
>
> 22. Eq. 30: r are consecutive recall levels and $$p_{\text{interp}}(r) = \max_{\tilde{r} \ge r} p(\tilde{r})$$
>  is interpolated precision, following standard PASCAL VOC metrics.
>
> 23. Training Protocol follows MapTRv2: ImageNet-pretrained backbone; all other modules trained from scratch.
>
> 24. Backbone Comparison (Swin-T):
> ResNet-50 was chosen as the standard benchmark to control variables. To ensure rigor, we added Swin-T results for MapUQ:
>
> | Method       | Backbone | Epoch | ped. | div. | bou. | mean |
> |--------------|----------|-------|------|------|------|------|
> | BeMapNet     | SwinT    | 30   | 64.4| 61.2| 61.7| 62.4|
> | PivotNet     | SwinT    | 30   | 63.8| 58.7| 64.9| 62.5|
> | MapUQ (Ours) | SwinT    | 30   | 64.4| 61.5| 65.2| 63.7|
>
> MapUQ (ResNet-50) at 69.7 AP already surpasses SOTA MapVR (Swin-T) at 66.8 AP, proving the efficacy of our UQ method regardless of backbone.
>
> 25. Deep Ensembles are computationally prohibitive for real-time industrial mapping, as costs scale linearly with ensemble size. MapUQ achieves uncertainty-awareness with minimal cost via single-pass feature extraction.
>
> Regarding the hyperparameter settings, please refer to our response to Reviewer 3.

---

> > ### Author Rebuttal · Reviewer_hBki · 2026-04-01
> >
> > Thank you for mostly resolving the minor remarks. What about the weaknesses section? This seems to be left out with the exception of the hyperparameter remark.

---

> > > ### Author Response · Authors · 2026-04-02
> > >
> > > Due to space limitations, we prioritize addressing the Detailed Remarks. Our responses to Weaknesses (excluding hyperparameters) are as follows:
> > >
> > > 1.Our lab focuses on Uncertainty Quantification (UQ),applied here to challenging end-to-end BEV tasks.MapUQ’s modules are highly generalizable:MSUH suits any semantic segmentation;PRSA’s uncertainty-based adjustment extends to point-output tasks like pose estimation or medical imaging;and NSC scales to LLMs and noisy label learning.Our approach is a principled instantiation of universal UQ theory with broad cross-domain applicability.
> > >
> > > 2.We believe the mention of mIoU is a misunderstanding,as neither our paper nor standard BEV vectorized mapping literature utilizes this metric.Pixel-based mIoU fails to reflect the geometric and topological accuracy of vectorized instances;thus, distance-based  mAP is the de facto standard, where MapUQ achieves significant gains.Unlike mIoU,which is often diluted by simple samples,MapUQ’s value lies in high-risk scenarios and reducing Expected Calibration Error (ECE). While we have achieved competitive IoU in previous industrial frameworks with complex lane types,it remains unnecessary for nuScenes/AV2 and consistent with SOTA BEV research.
> > >
> > > 3.We appreciate the insightful question regarding calibration.To clarify:
> > > Temperature Parameter: The $\tau=1.5$ in Sec 4.3 (Table 8) is a hyperparameter for the PRSA module (Eq.10) to modulate geometric ROI scaling based on error proxies.It is not used for post-hoc temperature scaling of classification scores.
> > > Intrinsic Calibration: MapUQ achieves superior results (lower ECE/NLL in Table 4) through uncertainty-aware training objectives,not post-hoc techniques like Platt or Temperature Scaling. Unlike post-hoc methods that merely rescale outputs, our modules fundamentally reshape the feature space and confidence distribution during training.For a fair comparison, no post-hoc re-calibration was applied to MapUQ or the baseline (MapTRv2).During evaluation, Dropout was enabled for both to ensure stochastic consistency.Thus,Table 4 reflects each architecture's intrinsic calibration capability. While scaling can lower ECE,it cannot resolve confidence mis-ordering or hallucinations in occluded regions;our NSC module explicitly penalizes these during learning for greater robustness. We will clarify these distinctions in the revised manuscript.
> > >
> > > 4.Our architecture is grounded in Bayesian DL and variational inference. MC Dropout approximates weight posteriors [Gal et al., ICML'16]. Multi-task consistency loss minimizes KL divergence [Kendall et al., CVPR'18], and uncertainty-based regularization acts as adaptive weighted likelihood estimation [Chang et al., ICCV'19]. These reflect established statistical principles, not ad-hoc engineering.
> > >
> > > 5.Regarding the evaluation datasets, we clarify that all experiments for MapUQ were conducted strictly using the official nuScenes V1.0 and Argoverse 2(AV2) datasets. For nuScenes, we followed the standard protocol (700 training / 150 validation scenes) with a perception range of 30m forward/backward and 15m left/right.For AV2,we utilized the standard sensor configurations and data partitions to verify our model’s robustness in more challenging,long-range scenarios.Across both datasets,we performed vectorized evaluations on three core map elements: pedestrian crossings,lane dividers,and road boundaries.Our metric calculations strictly adhere to the mAP protocol based on point-to-line distance thresholds at 0.5m,1.0m,and1.5m.We have explicitly labeled the respective data source (nuScenes or AV2) in each experimental table in the manuscript.
> > >
> > > 6.While Eq.1-7(global encoding) and Eq.23-29(NSC) share operator structures, their inputs, distributions, and objectives differ. To address the redundancy concern, we will restructure Eq.23-29 as a generalized extension of Eq.1-7 in the revision,clearly denoting variable substitutions to maintain rigor while ensuring conciseness.
> > >
> > > 7.We sincerely thank the reviewer for the feedback on readability.We will restructure Figure 1 with hierarchical color-coding and optimized flows to clarify the BEV mapping pipeline. For Figures 2&3, we clarify their technical logic: Figure 2 (PRSA): It captures real-time prediction quality by measuring geometric residuals between predicted points and ground truth. This deviation is defined as a spatial uncertainty signal, which dynamically adjusts the ROI sampling radius for subsequent decoder layers.This ensures the receptive field adapts to the target’s geometric manifold, preventing ineffective refinement in erroneous regions.Figure 3 (NSC): It details the sample screening logic.Tokens from the Map Decoder undergo stochastic inference via Monte Carlo Dropout (MCD) to generate uncertainty-aware scores. These scores guide the selection of positive/negative samples to calculate $L_{pos}$, $L_{union}$, and $L_{entropy}$, effectively leveraging uncertainty to learn from challenging negative samples.

---

### Official Review · Reviewer_ftUn · 2026-03-13

**Soundness:** 3
**Presentation:** 3
**Significance:** 3
**Originality:** 3
**Overall Recommendation:** 4
**Confidence:** 3

**Summary:**

The paper studies the integration of uncertainty quantification into end-to-end Bird's-Eye-View (BEV) vectorized map construction for autonomous driving. The authors propose MapUQ with: (1) the Multi-task Semantic Uncertainty Head (MSUH), that uses uncertainty estimates to re-weight segmentation losses; (2) the Progressive ROI Scale Adapter (PRSA), which dynamically modulates the deformable attention search range; and (3) the Negative Sample Classifier (NSC), which uses MC-Dropout-derived uncertainty scores for hard negative mining for lane classification. Extensive experiments are done on nuScenes and Argoverse 2.

**Compliance With Llm Reviewing Policy:**

Affirmed.

**Final Justification:**

The authors addressed the concerns in the rebuttal.

**Key Questions For Authors:**

- To alternative to Eqn9, ablate the proxy design choices: compare (a) variance of repeated dropout predictions, (b) entropy of the predicted point distribution, and others. Evaluate each on both AP and ECE metrics.

**Limitations:**

Yes

**Strengths And Weaknesses:**

Strengths
- Interesting problem: The paper picked uncertainty quantification problem in AV in three concrete failure modes i.e. misclassification, spatial drift, and semantic segmentation errors and each mapping to a specific module, which is a clear and well-structured framing. The uncertainty visualization results (Figures 6, 9) are genuinely compelling
- The paper demonstrate a negligible overhead time which is apt for AV settings.
- Extensive experiments and ablation studies are performed on two datasets: nuScenes and Argoverse 2.

Weaknesses
- Dependence on MC-Dropout: two components of MapUQ i.e. MSUH and NSC rely on $T$ stochastic MC-Dropout forward passes during both training and inference. However, traditional UQ methods like MC-Dropout were criticized for this multiple forward passes during inference which is not ideal for AV settings.

- The output head costs only 0.8 ms. Are stochastic forward passes are performed through the classification head with each query? Wont this linearly scale with increase in $T$. It will be nice to report the exact value of $T$ used for each module (MSUH, NSC) at inference time. And, plot mAP and ECE versus T to find the right point and justify the chosen T.

- Is MapTRv2 baseline run with $T$ forward passes? This can provide good comparison of its runtime and calibration against MapUQ with the same T.

- NSC module selects negative samples as predictions with high epistemic uncertainty. How does that justify? Because, high epistemic uncertainty need not mean that the sample is a hard negative but it may simply be an uncertain correct prediction in a genuinely ambiguous scene region.

- Continuing the earlier point, it will be nice to have analysis on the composition of the mined negative sample set. We can report what fraction of high-uncertainty predictions are (a) correctly classified but uncertain, (b) misclassified with high uncertainty, and (c) spurious detections. This would validate whether Uncertainty is a good proxy for hard negatives.

- In L176, the paper states it uses 'the dispersion of the predicted point sequence as a proxy metric for uncertainty.' at inference time. However, this proxy is not formally defined, and no ablation is provided for comparing the training-time error signal against the inference-time proxy.

- Missing ablations: Ablating the use of aleatoric vs. epistemic uncertainty separately in both MSUH and NSC and reporting AP, ECE, and NLL would justify the joint use of both components in Eqn 3.

Minor
- Captions of figures and tables can be self-explanatory

---

> ### Author Rebuttal · Authors · 2026-03-30
>
> Dear Reviewer,
> Thank you for your insightful comments. We respond to your concerns as follows:
> (1) Efficiency of MC-Dropout and choice of T.
> Although MC-Dropout is often considered computationally expensive, we restrict stochastic sampling only to the lightweight output head, rather than the full model. Specifically, we use T=5 (MSUH) and T=20 (NSC). The output head accounts for only 0.8 ms out of 71.9 ms (~1.12%), making the overhead negligible. For additional details , please refer to our response to Reviewer vEQ7
> We further validate the choice of T:
>
> | T  | mAP | ECE |
> |----|-----|-----|
> | 10 | 62.2 | 10.4 |
> | 15 | 62.6 | 9.8  |
> | 20 | 62.9 | 9.3  |
> | 25 | 62.9 | 9.2  |
> | 30 | 63.0 | 9.2  |
>
> Performance saturates at T=20, achieving a good balance between efficiency and accuracy.
> (2) Fairness of comparison.
> To ensure fairness, we also enable Dropout in the MapTRv2 baseline output head, ensuring consistent uncertainty estimation across methods.
> (3) Rationality of NSC.
> We use uncertainty as a proxy for potential hallucinations, which often occur in OOD regions (e.g., occlusion, low light). NSC does not suppress “uncertain but correct” predictions; instead, it refines the decision boundary by incorporating high-uncertainty samples as hard negatives, effectively reducing false positives.
> A component analysis on 500 high-uncertainty samples shows:
>
> | Type                     | Ratio |
> |--------------------------|-------|
> | Correct but uncertain    | 8.6%  |
> | Misclassified            | 55.2% |
> | Spurious detections      | 36.2% |
>
> Thus, 91.4% correspond to actual failures, validating epistemic uncertainty as a reliable indicator. Even the remaining 8.6% act as useful regularization, preventing overconfidence.
> (4) PRSA proxy (Dispersion).
> We define Dispersion as the variance of predictions from T stochastic passes. Correlation analysis shows a strong Pearson correlation (r = 0.83) with L2 error, validating it as an effective proxy.
>
> | Version   | Strategy      | mAP  | ECE  |
> |-----------|--------------|------|------|
> | Baseline  | Fixed scale  | 61.5 | 13.1 |
> | Random    | Random scale | 60.8 | 13.5 |
> | Oracle    | GT error     | 63.8 | 8.9  |
> | Proposed  | Dispersion   | 63.0 | 9.3  |
>
> Our method is only 0.8% below Oracle, showing Dispersion effectively approximates true error.
> (5) Aleatoric vs. Epistemic uncertainty (MSUH & NSC).
> We conduct ablations to analyze their roles:
>
> | MSUH Config | mAP  | ECE  | NLL  |
> |-------------|------|------|------|
> | Baseline    | 61.5 | 13.1 | 2.89 |
> | Aleatoric   | 62.1 | 9.8  | 2.15 |
> | Epistemic   | 62.7 | 11.2 | 2.54 |
> | Both        | 63.0 | 9.3  | 1.96 |
>
> Aleatoric → handles noise, improves calibration (ECE ↓)
> Epistemic → handles knowledge gaps, improves accuracy (mAP ↑)
> Joint modeling achieves Pareto optimality
> For NSC:
>
> | Config      | mAP  | ECE  |
> |-------------|------|------|
> | Only ℒcls   | 61.5 | 13.1 |
> | Aleatoric   | 61.6 | 12.8 |
> | Epistemic   | 62.8 | 10.5 |
> | Both        | 63.0 | 9.3  |
>
> Epistemic uncertainty is dominant in hard negative mining, while aleatoric plays a limited role due to redundancy with MSUH.
> （6）Alternative proxies for Eqn. (9).
> We thank the reviewer for the insightful suggestion regarding alternative proxy designs for Eqn. (9). In response, we conduct a comprehensive comparative study on three candidates: (a) coordinate variance, (b) prediction entropy, and (c) inter-layer refinement magnitude. The results reveal distinct characteristics of each proxy:
> Inter-layer refinement magnitude: the L2 distance between the predicted points of the current decoder layer and those of the previous decoder layer.
> The experiments show that inter-layer refinement magnitude (LRM) achieves the best performance in terms of mAP, as it directly provides a feedback signal for geometric correction. However, due to the lack of stochastic distribution information, it performs the worst on the ECE metric. In contrast, prediction entropy (Entropy) can significantly improve ECE, but since it is insensitive to geometric spatial deviations, its contribution to mAP is limited.
> Our method—coordinate variance—achieves highly competitive second-best performance on both mAP and ECE. From a theoretical perspective, coordinate variance, obtained via MC-Dropout sampling, captures both the geometric variability in spatial predictions and the model’s epistemic uncertainty.
> Considering that autonomous driving tasks require both mapping accuracy (mAP) and prediction reliability (ECE), coordinate variance provides an optimal trade-off. It not only effectively guides PRSA for receptive field adjustment (i.e., correcting geometric errors), but also offers reliable confidence estimation.
> For more details on hyperparameter settings, generalization experiments, and additional extended studies, please refer to our response to Reviewer 4755.

---

> > ### Author Rebuttal · Reviewer_ftUn · 2026-04-04
> >
> > Thanks authors for addressing the concerns. I wish to raise my scores.

---

> > > ### Author Response · Authors · 2026-04-04
> > >
> > > Thank you for your recognition of our work and for the increase in the score of our paper. May I politely ask how many points you plan to raise it by? Could you please change the score?

---

### Decision · Program_Chairs · 2026-04-30

**Decision:**

Accept (regular)

**Comment:**

Four reviewers gave overall positive scores: three ‘Weak accept’, and one ‘Weak reject’. During the rebuttal phase, the authors successfully addressed almost all the concerns raised by the reviewers. Based on all of these, the decision is to recommend the paper for acceptance. However, it is recommended that the authors revise the paper according to the reviewers’ comments when the paper is finally accepted.